Biological science practices  

ecology, environmental science

anti-racism, BIPOC, colonialism, equity, diversity, inclusion

# Overcoming racism in the twin spheres of conservation science and practice

Lauren F. Rudd[1,2,†], Shorna Allred[4,5], Julius G. Bright Ross[1,2], Darragh Hare[1,2,4], Merlyn Nomusa Nkomo[6], Kartik Shanker[7,8], Tanesha Allen[1], Duan Biggs[9], Amy Dickman[1,2,10], Michael Dunaway[11], Ritwick Ghosh[12], Nicole Thompson González[13], Thembela Kepe[14,15], Moreangels M. Mbizah[16,17,18], Sara L. Middleton[1,3], Meera Anna Oommen[8], Kumar Paudel[19,20], Claudio Sillero-Zubiri[1,2,21] and Andrea Dávalos[22]

[1]Department of Zoology, [2]Wildlife Conservation Research Unit, and [3]Department of Plant Sciences, Oxford University, UK
[4]Center for Conservation Social Sciences, Department of Natural Resources and the Environment, and
[5]Department of Global Development, Cornell University, USA
[6]Fitzpatrick Institute of African Ornithology, University of Cape Town South Africa, Cape Town, South Africa
[7]Centre for Ecological Sciences, Indian Institute of Science, India
[8]Dakshin Foundation, India
[9]Environmental Futures Research Institute, Griffith University, Australia
[10]Lion Landscapes, Tanzania
[11]Department of Sociology, Syracuse University, USA
[12]Global Futures Laboratory, Arizona State University, USA
[13]Department of Anthropology, University of New Mexico, USA
[14]Department of Geography, University of Toronto, Canada
[15]Geography Department, Rhodes University, South Africa
[16]Wildlife Conservation Action, Zimbabwe
[17]Department of Evolutionary Biology and Environmental Studies, University of Zurich, Switzerland
[18]Sustainability Research Unit, Nelson Mandela University, George, South Africa
[19]Greenhood Nepal, Nepal
[20]Department of Geography, University of Cambridge, UK
[21]Born Free Foundation, Ethiopia and UK
[22]Biological Sciences Department, SUNY Cortland, USA

LFR, 0000-0002-5925-9514; SA, 0000-0001-6237-0638; JGBR, 0000-0003-2454-1592; DH, 0000-0003-4418-9637; MNN, 0000-0002-0609-0153; KS, 0000-0003-4856-0093; NTG, 0000-0002-3195-1277; TK, 0000-0002-1807-0387; MMM, 0000-0002-6868-052X; SLM, 0000-0001-5307-8029; AD, 0000-0002-3590-4152

It is time to acknowledge and overcome conservation's deep-seated systemic racism, which has historically marginalized Black, Indigenous and people of colour (BIPOC) communities and continues to do so. We describe how the mutually reinforcing 'twin spheres' of conservation science and conservation practice perpetuate this systemic racism. We trace how institutional structures in conservation science (e.g. degree programmes, support and advancement opportunities, course syllabuses) can systematically produce conservation graduates with partial and problematic conceptions of conservation's history and contemporary purposes. Many of these graduates go on to work in conservation practice, reproducing conservation's colonial history by contributing to programmes based on outmoded conservation models that disproportionately harm rural BIPOC communities and further restrict access and inclusion for BIPOC conservationists. We provide practical, actionable proposals for breaking vicious cycles of racism in the system of conservation we have with virtuous cycles of inclusion, equality, equity and participation in the system of conservation we want.

**Author for correspondence:**
Lauren F. Rudd
e-mail: lauren.rudd@gtc.ox.ac.uk

[†]Present address: Green Templeton College, 43 Woodstock Road, Oxford, OX2 6HG, UK.

# 1. Introduction

It is time to acknowledge and overcome conservation's deep-seated systemic racism, which has historically marginalized Black, Indigenous and people of colour (BIPOC) communities, and continues to do so [1–5]. Given conservation's history of racism, exclusion and oppression [1,6], and the fundamental role that BIPOC communities must play in biodiversity conservation, conservation researchers and practitioners must lead the way in committing to anti-racism [7]. Failing to examine, acknowledge and act on persistent oppression in our field provides tacit support to racism, tarnishing the conservation successes we achieve, and causing real harm to some of the world's most vulnerable people [8].

In this paper, we critique contemporary mainstream conservation: formalized, evidence-based efforts to conserve biodiversity. Despite its relatively brief history, this form of conservation globally dominates indigenous knowledge systems through which people have actively and adaptively conserved ecosystems for millennia [9]. In many places, mainstream conservation has replaced indigenous knowledge systems, often to the detriment of local people and biodiversity [10].

We draw on existing literature and our interdisciplinary, cross-sectoral, professional experiences to identify issues of and propose solutions to systemic racism in what we term the 'twin spheres' of conservation: (i) 'conservation science': academic teaching and research, which typically takes place on college and university campuses and (ii) 'conservation practice': applied conservation policies and programmes, which typically take place outside the campus gates. We argue that systemic racism mars our activities in these twin spheres of conservation science and practice, and that what we do in each sphere affects what happens in the other (figure 1). Conservation practice's colonial origins and racist history influence how academia conceptualizes conservation problems and solutions, what is taught, and the nature of interactions between students, colleagues and the local people on whose land research is conducted [1]. The racism that permeates the academic sphere is reproduced in conservation activities outside academia, in the biases and preconceptions that conservation graduates carry with them and apply to on-the-ground decision making in the organizations for which they work. In turn, these on-the-ground decisions affect conservation practice by influencing which conservation problems are addressed, how they are addressed and how colleagues and collaborators are treated. Conservationists' formal and informal practices can, often implicitly or unintentionally, impart racist and neo-colonialist undertones onto academic work (e.g. publications, conference presentations, teaching) which underpins many of the conservation programmes and policies that are studied and taught to subsequent cohorts of students (figure 1).

This vicious cycle in the twin spheres of conservation science and practice characterizes the conservation we have created and inherited, but it does not characterize the conservation we want. We urge fellow members of the conservation community—academics and practitioners—to take stock of the manner in which much of what we do in conservation science and practice perpetuates, reinforces and deepens racial divisions [1], and reflect honestly on how we can change our behaviours and institutions for the better. We are morally obliged to demolish racist structures, reform our individual and collective actions, and construct a more equal, inclusive and socially just field. We owe this to the people whom conservation has harmed and continues to harm, the communities on whose land we are privileged to work, the students we mentor and the broader societies we serve. We recognize and strongly welcome many recent steps in the right direction, but we have a long way to go.

We, the authors, are a diverse team representing, other than race, different ethnicities, levels in academia, years of experience in both conservation research and practice, primary fields of study and specializations in conservation, organizational affiliations and regions of the world. While we do not purport to speak for all conservationists in our different communities, the varied perspectives we discuss here represent our lived experiences and are not appropriated knowledge.

We acknowledge that the experiences of BIPOC individuals in conservation will depend on political, social and economic factors such as nationality, native language and socio-economic status. While the extent of racism faced by individuals and the obstacles they encounter may vary, BIPOC individuals are, on the whole, a minority within the conservation space. We also acknowledge that the state of conservation varies around the world. In some previously colonized countries, white western organizations, individualism and ideals still largely dominate conservation, but in other countries, local and regional efforts predominate. Even where BIPOC individuals currently lead conservation research and practice, these individuals often seem to come from positions of relative privilege within society, regardless of whether the society is BIPOC majority or BIPOC minority. To truly reconcile the historic racial injustices within the field of conservation, this type of privilege needs to be acknowledged and addressed.

Without recognizing barriers to individuals such as socio-economic background, language, access to training and networking opportunities, simply increasing the representation of BIPOC people in conservation will not solve the problem, as being a BIPOC individual does not guarantee either cultural literacy or an anti-racist outlook. Racist hierarchies and processes operate within every society and at multiple levels, not simply at the global scale of colonial legacy. While much of this conversation is outwith the scope of this paper, what we advocate for above all is fostering greater inclusivity within conservation, which should go some way towards addressing all the problems outlined.

# 2. Conservation practice's deep-seated racist history

Many dominant conservation tools, such as protected areas and quotas for sustainable use, are rooted in colonial strategies for optimizing resource extraction and recreational opportunities on colonized land [11,12]. These practices came at great cost to local people, including through forced removal, abuse, loss of livelihoods, cultural assimilation, human rights abuses and death [13–15]. For example, Native American people were killed or forcibly removed from their ancestral lands to create national parks that appealed to settler colonists' wilderness ideals [13,16]. Long-standing indigenous and local cultural practices, norms and taboos were replaced by extractive or preservationist values of European colonists [9,17,18]. Contemporary conservation can perpetuate these values, often in spite of strenuous opposition from Indigenous and local people [19].

Proc. R. Soc. B 288: 20211871

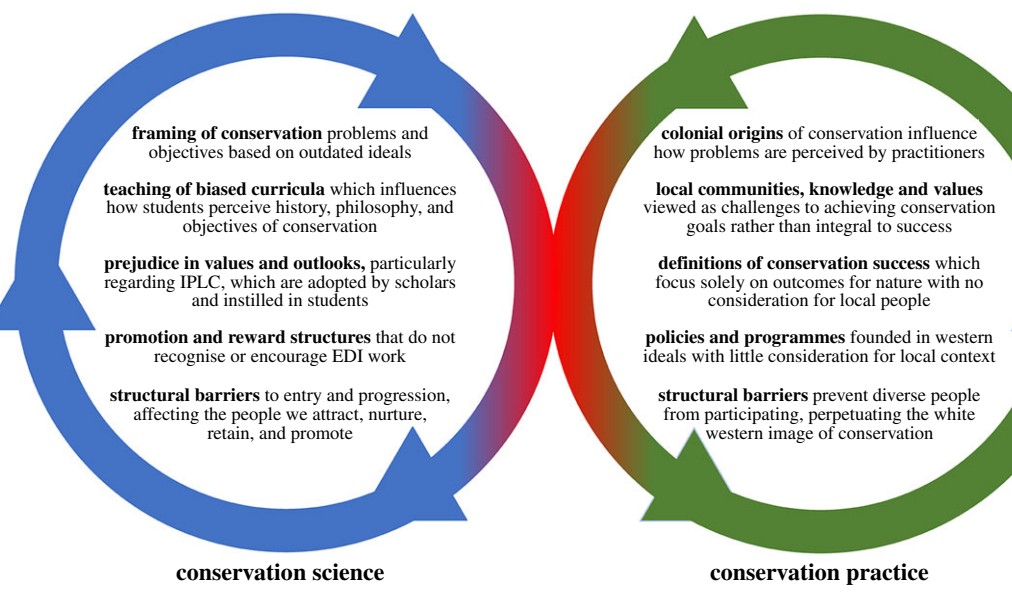

**Figure 1.** The mutually reinforcing twin spheres of conservation science and conservation practice. Although each sphere can operate largely independently of the other, they each perpetuate neo-colonial and racist ideologies that reinforce the other in subtle but important ways. Escaping this vicious cycle will require conservation scientists and practitioners to change our individual and collective behaviours (boxes 1–3). Definitions: Indigenous peoples and local communities (IPLCs); equality, diversity and inclusion (EDI). (Online version in colour.)

While extractive approaches can be clearly neo-colonial and racist, preservationism can be less obviously so. The preservationist approach seeks to preserve 'Eden'-like environments, often via protected areas [12,18]. Recognizing the extent of ecological degradation across much of the Global North, advocates of this approach appear to believe that conservation can only happen elsewhere in pristine environments, typically in the Global South. Preservationist efforts may be well-intentioned (e.g. by seeking to protect critically endangered populations or areas of high biodiversity) but are often blind to the environmental injustices they impose on local people through fortress conservation (conservation through formal, exclusionary protected areas, that displace and marginalize local people and prioritize the interests of wealthy, often distant, elites) [20]. In our experience, conservationists from the Global North often lack local cultural literacy and come equipped with the privileged legacy of colonial power, perpetuating a 'white saviour' mentality [21]. Related tensions are evident in 'parachute science', in which external conservationists suddenly arrive in a new place to conduct research, using local scientists only as field staff or data collectors under the pretext that local capacity or expertise is lacking [22,23]. Some of the best-known examples of conservation practice, as well as many academic conservation scientists' field experiences, are enmeshed in such unjust paradigms. When academics bring these examples and experiences uncritically into formal and informal teaching, conservation students may internalize them as normal or desirable.

Conservation's colonial underpinnings continue to permit practices that subjugate local people by portraying them as responsible for conservation problems, forcibly removing them from their land in the name of conservation and preventing them from accessing wildlife and protected areas [6,8,24], often by militarized means [25,26]. Some influential researchers and advocacy groups based in the Global North advocate for extending their preferred conservation ideologies to vastly different socioecological and cultural

contexts, with apparently little regard for traditional practices or ethics in those locations [15]. Such prescriptions can endorse social hierarchies (e.g. caste in India) by privileging certain practices (e.g. vegetarianism) without understanding the historical and social inequities associated with them [27]. More broadly, these practices disempower people in the Global South by demanding they change their behaviours, many of which they have been practising for millennia, to suit the preferences of distant interest groups. Such demands are particularly distasteful when couched, seemingly without irony, in anti-colonial and pro-equality rhetoric [28]. High-profile proposals to increase the amount of land and seascapes designated as exclusionary protected areas (e.g. [29]) show little consideration for social and cultural consequences [30]. Western interests claiming or maintaining de facto control over many conservation spaces in the Global South is straightforward neo-colonialism [31], a contemporary form of land grabbing permitted in the name of environmental protection [32].

## 3. Exclusion from engaging with nature

The high degree of exclusion of BIPOC people across levels of conservation science and practice reproduces conservation practice's colonial history. Many BIPOC people have been excluded from environmental policymaking. The ability for indigenous communities to effectively participate in policies that affect them has been removed through colonial processes in many parts of the world. For example, the Marshall Trilogy of Supreme Court decisions (Johnson v. M'Intosh 1823, Cherokee Nation v. Georgia 1831, and Worcester v. Georgia 1832) in the United States, the Treaty of Waitangi (1840) in New Zealand, and the policy of Terra Nullius in Australia (1835), all placed Indigenous sovereignty over land and resources within the dominion of colonial governments.

In BIPOC minority countries, people of colour are further excluded from conservation because they are less likely to

(a)

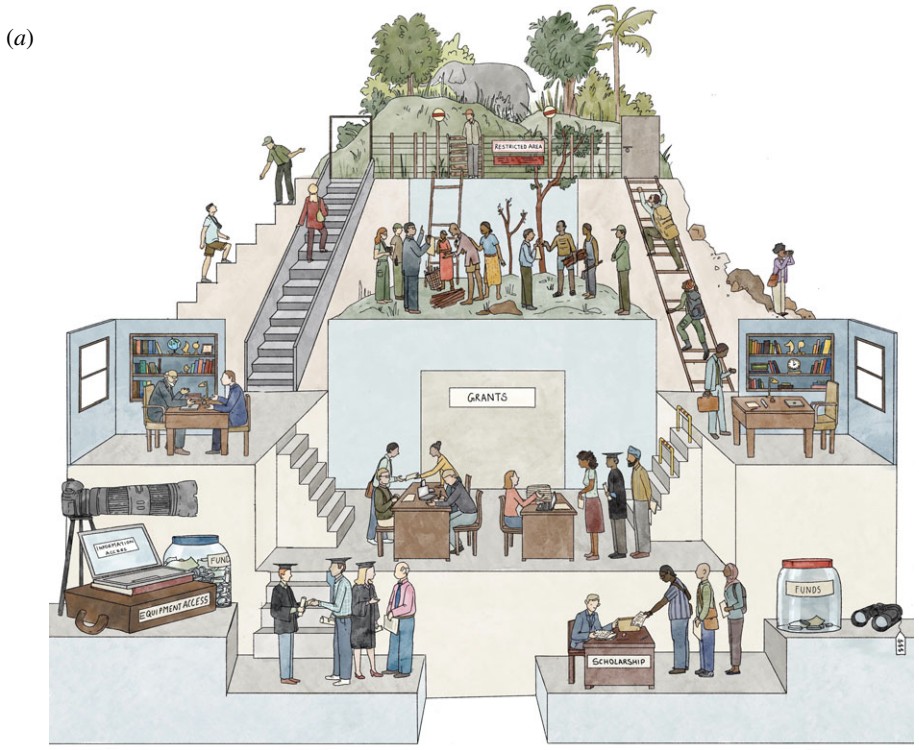

(b)

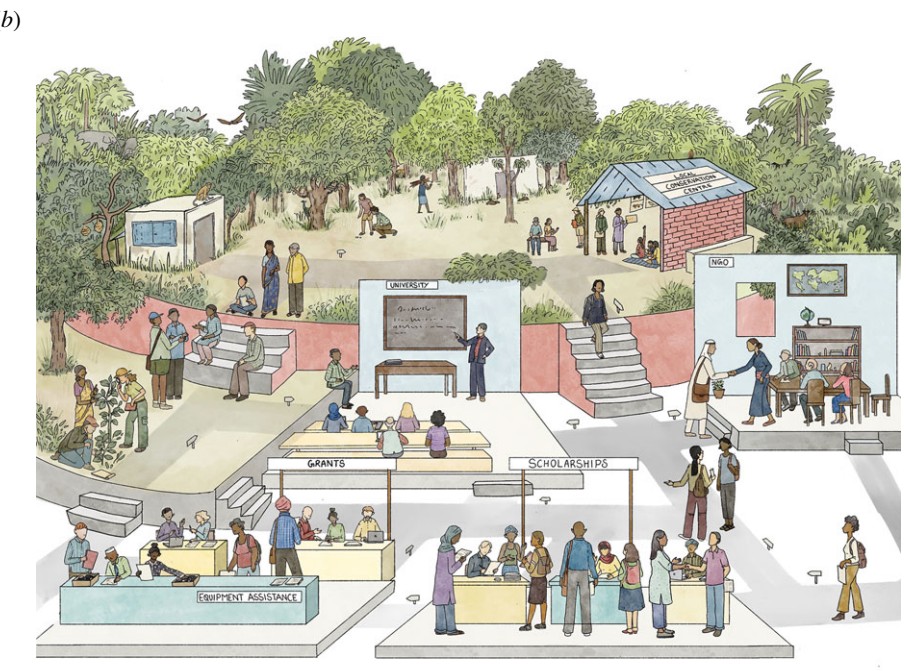

**Figure 2.** The conservation we have (*a*) and the conservation we want (*b*). Current pathways to success systematically favour some groups over others. Each step in the academic process represents a successive impediment to aspiring BIPOC conservationists, from the resources to pursue such a career, to the attentiveness of supervision received, to the degree of welcome that recent graduates of different skin colours receive in the industry. Consequently, conservation practice is designed and communicated to local people by outsiders who may fail to understand local context or are beholden to predominant western approaches to conservation. We must strive to bring about a system that is more attractive and more accessible to BIPOC aspirants. The academic system should only represent one valid entry point to conservation. By enabling the sharing of expertise from local conservationists and increasing career mobility between field conservation, academia and the non-profit sector, multiple stakeholder viewpoints can be prioritized in the process of moving towards more holistic, novel models of conservation. Illustration by Barkha Lohia. (Online version in colour.)

have access to and use outdoor spaces for recreational purposes than people from predominantly white communities [33,34]. Recent high-profile cases in the United States and United Kingdom demonstrate that people of colour, and

particularly Black people, are often unsafe and unwelcome in outdoor spaces [35,36].

Exclusion is also evident in financial hurdles to entering conservation, particularly for many from BIPOC communities

**Box 1.** How to recognize and address the unjust history of conservation science.

1. Educate oneself on the history of racism in conservation through reading, reflecting on one's own experiences and engaging in dialogue with others. Recommended reading [6,11–13,16,20,31].

2. Diversify and broaden the curriculum; teach a more comprehensive representation of past and present conservation practice, including the work and perspectives of BIPOC scholars, and ultimately produce new 'standard' textbooks that encompass this history. Recommended reading [2,59–63].

3. Prioritize inclusive teaching practices in conservation courses by embracing the tenets of inclusive course design, active learning modalities and service learning techniques, to encourage broader participation and interest in conservation sciences. Recommended reading [64–67].

4. Conduct outreach in predominantly BIPOC schools and areas within predominantly white countries to promote conservation careers at an early age. Potential outreach activities could include hands-on activities, 'meet a conservation scientist' Q&A session, talks at school career days and hosting research events tailored for high school students.

5. Encourage professional associations to fully integrate equity, diversity and inclusion (EDI) into their policies and standards, (for example, the Society for Conservation Biology should update their guidelines for conservation literacy to include a section on EDI).

6. Question dominant narratives about what works in conservation, including how success is measured, and track the history of power relations shaping such narratives.

7. Recognize that injustice is not only historic or organizational, but occurs today within each of our lives. Challenging conversations, personal reflection and honesty are required for each of us to take personal, immediate steps to ensure we are not perpetuating unjust actions.

8. Understand that facing past wrongs is not just about rehashing the past, but to be honest about the present and to create a solid foundation for future action.

with a long history of economic exclusion. As examples, activities such as birding, diving and hiking to name a few require equipment that is often costly [37]. This financial barrier can be further amplified in some areas of the Global South if such equipment is not locally available or affordable, making it difficult to source. Because experience builds passion for the outdoors, inspiring people to pursue careers in conservation, socio-economic barriers block many BIPOC people's routes into conservation science and practice. Exclusion can be concentrated in certain sub-communities: individuals of different genders, migration experience and wealth within a recognized ethnic minority group may vary widely in their motivations and ability to use and interact with outdoor spaces [38]. BIPOC members of other minority communities may face additional barriers to safely using outdoor spaces [39,40]—experiences which can be compounded by racism. Further, colourism exacerbates the threat to dark-skinned BIPOC individuals, and the discrimination they face [41]. These converging forms of discrimination illustrate the magnitude and diversity of obstacles that systematically divert BIPOC people away from conservation (figure 2).

## 4. Racism in conservation science and practice today

Power in conservation typically resides in governments, corporations, large NGOs and universities [42]. Universities are central because they provide the qualifications required for a degree in conservation. However, BIPOC students are disproportionately underrepresented in degree subjects that lead to conservation careers [5,43,44], partly due to high upfront degree costs and lack of scholarships, expensive field trips, unpaid summer field experiences, low job security

and the predominance of low-paid or voluntary entry-level positions [45]. Once enrolled, students are expected to undertake conservation work during summers and academic holidays to boost their credentials. However, field-based educational experiences may not always be designed with inclusivity in mind [46] and can perpetuate neo-colonial attitudes when being run by institutions outside the host country [47], making the experiences unsafe and uncomfortable for BIPOC students. Internship opportunities in universities, NGOs, governmental and intergovernmental agencies often target students from wealthier countries and are typically both expensive to enrol in and unpaid, thus carrying substantial transaction and opportunity costs [48,49]. Many BIPOC students cannot participate for financial or cultural reasons, missing out on valuable work experience, networks and job opportunities. Expectations that students hoping to work in conservation should go above and beyond and not expect a financial reward for their efforts, excludes many.

The predominant narrative of conservation taught in academia uncritically emphasizes the Western paradigm of pristine wilderness and fortress conservation, what Shanker & Oomen [31] term 'pristianity' named for the religious zeal in which preservation of wild spaces is pursued. Students in conservation degrees typically do not learn the colonial and deeply racist intentions and consequences of fortress conservation. Local knowledge is often referred to in passing as 'indigenous knowledge systems', relegating it to superstition and alternative thought while western ideas are imposed as the only way of understanding or engaging with ecosystems. Teaching this sanitized history of conservation perpetuates deep inequalities in the field and can alienate BIPOC students [50].

Advanced degrees are essential for many high-level conservation jobs, but funding for postgraduate study is scarce and predominantly flows to white students. For example, in

**Box 2.** How to construct better ways to conduct research and practice conservation.

1. Develop qualifying assessments for individuals to demonstrate 'cultural literacy' in relation to fieldwork sites (deliverables could be to incorporate local history, expected socio-economic impact, plans for local collaboration and plans for preventing neo-colonial relations in project proposals).

2. Ensure fair dissemination of funds and grants to BIPOC academics, conservation practitioners and BIPOC led organizations.

3. Develop new models to ensure that BIPOC voices are heard: e.g. balancing expensive, in-person networking events with opportunities for online networking (while being considerate of any technological barriers), to enable more participation from diverse conservationists.

4. Avoid 'parachute science'; meaningfully include local partners in conservation from question formulation and applied practice all the way through to publication and beyond (this applies to academic research and work done by NGOs and government agencies). For example, journals and funding bodies could require inclusion of local partners as co-authors or require a report of actions implemented to ensure inclusivity and equity when conducting research abroad [68].

5. Create opportunities for community members to have real agency in conservation projects and promote conservation management solutions that align with the communities' culture and values, even when those might conflict with the views of NGOs or other external stakeholders.

6. Respect the rights of Indigenous People and local communities to manage, benefit from, and sustainably use their resources, embracing—not suppressing—diverse conservation ethics and resource management systems.

7. Recognize that BIPOC communities are diverse and heterogeneous and have different values and cultures.

8. Promote bottom-up conservation practices that decentralize management practices and decision making. To do this, practitioners should embrace the core concepts from participatory action research, community-based research and indigenous methodologies, all of which focus on rebalancing power dynamics [69–75].

9. Collaborate with colleagues in history, political ecology, geography and other cognate disciplines to ensure inclusion of a broader perspective within conservation curricula, and that we consider critical perspectives throughout research design and implementation.

---

**Box 3.** How to create an inclusive, safe conservation that welcomes BIPOC individuals and allows them to thrive in conservation science and practice.

1. Ensure that 'essential' work experience is incorporated into undergraduate and graduate conservation degree programmes, is fully funded at this stage and is not used as a metric to judge candidates during admissions processes to these programmes, as they do not represent candidates' abilities but rather their opportunities.

2. Evaluate current harassment reporting and risk assessment procedures to ensure they protect anonymity and allow for reporting of issues specific to BIPOC individuals.

3. Recognize and reward EDI work in the same way we would traditional academic achievements.

4. Advocate for and actively create opportunities for your BIPOC colleagues, even when this means personally stepping aside/turning down opportunities.

5. Extend current EDI initiatives (e.g. for gender equality) to be inclusive of BIPOC individuals who also fall within the remit [76].

6. Protect BIPOC people in your team—learn through independent research and training programmes how, where and why they may be vulnerable. Listen with humility and compassion to their expressed concerns. Further, prevent their possible harm by, for example, creating a risk management plan for fieldwork, including mitigating strategies [39].

7. Learn the cultural histories, norms and values of the communities on whose land you conduct research, and incorporate them into your conservation. Include local people as partners to help define, measure and evaluate success.

---

the 2018–2019 application cycle for postgraduate study funding in the UK, only 6% of Natural Environment Research Council (NERC) studentships were awarded to ethnic minorities [51]. In the UK, success for white principal investigators applying to NERC for funding awards was 13% higher than for individuals from ethnic minorities [51]. Senior positions in environmental organizations are typically held by white people: as of 2014, people from minoritized ethnic groups occupied less than 12% of these leadership positions in the US [52]. Insular hiring practices such as advertising positions internally and developing unpaid internships into paid positions or degree scholarships exacerbate the problem.

Postgraduate study can be daunting, particularly to first-generation students, and BIPOC students disproportionately fall into this category [53]. In the light of this, respectful, supportive relationships between postgraduate students and

their supervisors are pivotal to success. People generally prefer to work with those that they can relate to and have a common culture with—a concept known as 'affinity bias' [54], which can be compounded by colourism [55], and further disadvantage BIPOC students in a field dominated by white people. Costs of attending international conferences and publishing scientific articles, which are both crucial for career advancement, can pose prohibitive financial barriers. Such factors are compounded by additional barriers such as visa processes and expenses to exclude people from BIPOC-majority countries from studying overseas [56].

This series of obstacles (figure 2) to success means BIPOC researchers are woefully underrepresented in conservation science and practice, and those who remain have few opportunities for advancement. Lack of high-level representation means little consideration is given to the specific problems that BIPOC people encounter in trying to succeed within conservation. We have personally witnessed or experienced many of these problems in our own workplaces. We have observed how the unique welfare and safety challenges to BIPOC conservationists, both in the field and in the workplace [39], can be invisible to senior colleagues who are unaffected by them and cannot relate. Racial stereotypes and derogatory language are used too often when discussing local communities and field staff, which alongside relentless assumptions about where one is 'really from' when referring to colleagues of colour, further alienates BIPOC people. Further, 'old-boy' networks and low turnover of individuals in senior positions mean that encountering racism and discrimination remains common [50]. The burden of calling out and reporting such incidents often falls on BIPOC people, which is especially daunting to those in junior positions because harassment reporting procedures in organizations with few possible BIPOC complainants cannot guarantee anonymity.

Academia is not the only route into conservation, but it currently acts as the main gatekeeper. Other entry forms (such as on-the-ground experience, often held by local conservation workers) may actually equip people with many more useful skills and fewer harmful biases. However, broadly speaking from our collective experience, a lack of academic qualifications (sometimes compounded by language barriers) prevents people from being able to progress to higher level positions in organizations where decision-making power resides. Among development fields, conservation appears to have an almost 'Brahminical' reverence for academic qualifications. As such, and despite a greater emphasis on BIPOC people and communities in the last two decades, conservation narratives remain dominated by western and/or privileged biologists and elite international and local NGOs [31,42].

## 5. Building inclusive conservation science and practice

Diversifying conservation has both ethical and practical consequences; it is socially just and can improve the success of conservation initiatives. It is important for conservation scientists and practitioners to acknowledge that, historically, BIPOC communities most impacted by environmental issues have been the least included in decision making [10]. Continuing to perpetuate these unjust power dynamics will wreak havoc on some of the world's most vulnerable communities [8]. Legitimate participation of local people produces better conservation outcomes because it builds community capacity and provides the opportunity for members to be involved in the definition of the problem, the development of policies and the implementation of measures and evaluation, ultimately increasing project sustainability [57]. Diversifying conservation teams increases the breadth of perspectives, driving innovation [58]—innovations that are sorely needed for developing ecologically and socio-culturally sustainable conservation strategies. Most importantly, respecting the rights of Indigenous People and local communities is required by international law, and as such is an imperative, not an optional luxury [8].

It is incumbent on all members of the conservation community to recognize and address the unjust history of conservation (box 1). For example, we must recognize that some of the earliest proponents of environmental protection in the Global North and South were also ardent proponents of colonial expansion, eugenics and white supremacy. We must acknowledge this context while dismantling it and seeking solutions rooted in a system of inclusivity and equality. It is essential that we reflect on the ways in which we have personally harmed or disadvantaged people from BIPOC communities in our professional lives. Holding ourselves accountable and taking steps to rectify these wrongs is a vital first step towards creating a more inclusive and just conservation. This sense of individual responsibility should be the basis for building future conservation solutions.

Many people from BIPOC communities are interested in conservation but are often excluded and alienated from it due to historic, unequal power and privilege structures. These structures must fall. It is therefore essential that all members of the conservation community play an active role in replacing the conservation we have with the conservation we want (figure 2). This means rethinking our individual and collective behaviours to create more inclusive institutions and organizations (box 2) and making conservation a field in which BIPOC communities can be safe and thrive (box 3).

## 6. Conclusion

Achieving excellence in conservation practice and promoting equity, diversity, inclusion and justice in conservation science are not mutually exclusive. Rather, they are all crucial to creating effective conservation practices that empower BIPOC communities by reforming our conservation institutions in both spheres. Conservation scientists who are also conservation practitioners are at the nexus of the twin spheres, and as such have both the greatest potential and responsibility to create positive change. We recognize that many individuals, organizations and groups are taking meaningful steps towards modes of conservation that empower BIPOC communities [1–4,59,60,77,78]. Nevertheless, there is still more to be done, and we must accelerate away from the exclusive and harmful institutions we have inherited, towards more inclusive and innovative institutions that promote conservation spaces in which people and nature thrive (figure 2). While some individuals have more power than others to affect change, every person can play a role in building conservation spaces that empower BIPOC communities.

We challenge all members of the conservation community, including ourselves, to use whatever privilege we have to make progress, however, small. We need to speak out against injustices, small or large, recruit those who are less privileged, promote them, give them a platform or step aside so they can

have ours. We need to change our syllabus and teach the difficult, shameful, aspects of conservation. We must acknowledge that some purported conservation successes come at an enormous and unconscionable cost to BIPOC communities and help prevent conservationists from committing similar errors in future. We need to strive to find that one inch of progress and then leverage it for systemic change. We must work within our spheres of influence to foster institutional change in research, practice, curricula, community partnerships, recruitment and retention, mentoring, and beyond.

Data accessibility. This article has no additional data.

Authors' contributions. L.F.R., T.A., J.G.B., A.Di., D.H., M.M. and M.N.N. conceived of the idea. L.F.R., S.A., J.G.B., A.Da., M.N.N., K.S. and D.H. drafted the manuscript. All authors revised manuscript drafts. All authors gave final approval for publication and agreed to be held accountable for the work performed therein.

Competing interests. We declare we have no competing interests.

Funding. This work was supported by the Natural Environmental Research Council (grant code: NE/L002612/1).

Acknowledgements. We thank Barkha Lohia for elegantly and skillfully transforming our words into beautiful illustrations.

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
