## [Peer Review File · Proceedings of the Royal Society B: Biological Sciences]

Review History

RSPB-2021-0374.R0 (Original submission)

Review form: Reviewer 1

Recommendation

Major revision is needed (please make suggestions in comments)

Scientific importance: Is the manuscript an original and important contribution to its field?

Acceptable

General interest: Is the paper of sufficient general interest?

Good

Quality of the paper: Is the overall quality of the paper suitable?

Acceptable

Is the length of the paper justified?

Yes

Should the paper be seen by a specialist statistical reviewer?

No

Do you have any concerns about statistical analyses in this paper? If so, please specify them explicitly in your report.

No

It is a condition of publication that authors make their supporting data, code and materials available - either as supplementary material or hosted in an external repository. Please rate, if applicable, the supporting data on the following criteria.

Is it accessible?

N/A

Is it clear?

N/A

Is it adequate?

N/A

Do you have any ethical concerns with this paper?

No

Comments to the Author

This is an impassioned plea for conservation to address issues of systemic racism. The authors focus on both the academic teaching and research environment as well as on the ground practice. The article is generally well-written and reasoned. However, there are a few ways that I think it could be improved.

1. Avoid over-generalizations. While it may seem that the only way to get the conservation community to pay attention is to make bold statements (and as a BIPOC researcher I totally understand the frustration behind the strength of these statements!), your largely non-BIPOC audience will actually dismiss the message if they feel that you are overgeneralizing. To give an example: "Racial stereotypes and often derogatory language used when discussing local communities and field assistants" (line 221) is flavoured by the word "often". Hang on, you don't present any survey or statistics to demonstrate that most conservation scientists use derogatory language most of the time when discussing local communities and assistants. Instead, I suspect that this statement is based on your personal experience, which of course is valid but nonetheless limited (too limited to use the word "often"). You could say, "In our experience, when racial stereotypes and derogatory language are used" or "Racial stereotypes and derogatory language are too often used". This is one example, but I felt this issue permeated the manuscript - I suggest tightening up the writing, carefully explaining what is the lived experience of the authors, and what are the conclusions of studies, and keeping the language used consistent with the source of the information would actually give this more punch in the end.

2. Stick tight to the evidence. For example, the authors write (line 159) "Hobbies such as birding and hiking require costly equipment [28]." While binoculars are definitely a cost barrier, I was intrigued what the costly hiking equipment might be, given that day hikes can be done in runners - so looked up the original article. The original article is actually about field courses, involving weeks of 8-10 hours a day in the field, and requiring sturdy hiking boots, backpacks, rain gear, etc. Again, if your audience thinks that you are playing loose with the facts, you will lose credibility. (as a sidebar, I think travel to hiking destinations and concerns about field safety is probably more of a barrier for casual hiking than footwear).

3. Provide concrete examples. While there are a number of useful examples presented here and there in the manuscript, I would have liked the ratio of examples to generalizations to be much higher. I do appreciate that you provide references for many of the broad statements you make, but the causal reader will not be familiar with all of these. Instead, providing a concrete example or statistic would be much more persuasive. For example, "These practices came at great cost to local people, including through forced removal, abuse, and murder [11]." (line 109). Murder! That may seem sensational to your audience. But if you used a sentence to outline an example from this reference that involved murder, then you'll be much more convincing.

4. Clarify relationship between BIPOC in Global North and Global South, and clearly differentiate the differences in their experiences of racism. Right now, there is little distinguishing between the unique experience of these two populations. And in fact, I felt that in some sections (like Exclusion from Engaging with Nature), the concerns of BIPOC in the North overtook the South. BIPOC in the North are dealing with being minorities in a Eurocentric and majority white culture. However, this doesn't mean that this experience makes us (I'll include myself) models of anti-oppression when it comes to engaging BIPOC in the South in conservation. Meanwhile BIPOC in the South have very different challenges - even when in a majority BIPOC country, there are intersectional issues (e.g. gender oppression, sexual orientation, access to economic privilege) that affect them both in terms of local engagement with the conservation community as well as attempting to gain academic credentials.

5. Restructure manuscript to clarify the "conservation science" vs. "conservation practice" division. In terms of the headings of sections, I was sometimes confused which we were talking about. I also did not find Figure 1 that useful: the points were vague, and it is never clear how conservation practice affects conservation science. You might instead consider a point form action plan for the conservation community that addresses the points you raise in text. Figure 2 was nicely done and engaging.

Line 48 - I believe it should be "syllabi" not "syllabuses"

Line 78 - define "traditional resource management systems". Are you referring exclusively to Indigenous resource management systems, and if not, what is the time frame and context for your definition?

Line 93. "preconceptions that conservationist graduates" instead of "preconceptions conservationist graduates"

Line 95. You describe "This vicious cycle" but above only describe conservation science -> conservation practice. For a cycle, you also need the arrow to go the other way.

Line 103. "people that conservation has harmed and continues to harm," instead of "people conservation has and continues to harm,"

Line 231 "hoFmbizping" ...?

Figure 1: please do not use acronym (IPLC) in figure with no explanation in legend.

Review form: Reviewer 2

Recommendation

Major revision is needed (please make suggestions in comments)

Scientific importance: Is the manuscript an original and important contribution to its field?

Good

General interest: Is the paper of sufficient general interest?

Excellent

Quality of the paper: Is the overall quality of the paper suitable?

Good

Is the length of the paper justified?

Yes

Should the paper be seen by a specialist statistical reviewer?

No

Do you have any concerns about statistical analyses in this paper? If so, please specify them explicitly in your report.

No

It is a condition of publication that authors make their supporting data, code and materials available - either as supplementary material or hosted in an external repository. Please rate, if applicable, the supporting data on the following criteria.

Is it accessible?

N/A

Is it clear?

N/A

Is it adequate?

N/A

Do you have any ethical concerns with this paper?

No

Comments to the Author

Review of Overcoming racism in the twin spheres of conservation science and practice
RSPB-2021-0374

We thank you for the opportunity to review this Letter. For context, three of us read the paper and we compiled our comments into one review. Our research expertise is broadly in ecology, evolution, and conservation. The three of us are also very engaged in equity, diversity, and inclusion work, although we are not scholars of critical race theory. We are a mix of males and females, BIPOC and non-BIPOC. We use I/We a bit loosely.

First and foremost, thank you for the time, effort, and emotional energy it took to write this piece. There is much injustice in the world but also there are many who have benefitted from the status quo and are reluctant to change. Although many of these types of commentaries have been written lately, there should be no limits on the number of times we say that change needs to happen. Having said that, our goal here is not to discount any of the authors' words or lived experiences, rather our goal is to try to provide feedback that strengthens what we feel are the core points of the paper.

General comments

The main points of this letter seem to be:

- (a) Black, Indigenous, and people of colour are underrepresented in conservation biology in the academy;
- (b) conventional western-led conservation practice is rooted in colonialism and racism; modern-day approaches to conservation (e.g. parachute science) ignore and disadvantage Indigenous voices and local communities (largely from the Global South).
- (c) Underrepresentation of BIPOC in conservation science perpetuates neocolonial and racist conservation practices, and vice versa.

Regarding point (a), several recent articles have highlighted that BIPOC are underrepresented in the academy, and especially in ecology-related fields. Some articles have focused on Academia in general (Barber et al 2020) while others have focused on ecology, evolution, and conservation (Chaudhury and Colla 2020, Massey et al 2021, O'Brien et al 2020, Wanelik et al 2020, Graves 2019). Because so many excellent papers have also been written on this topic, we are unclear as to how this current paper adds to the existing discourse. The reasons for why there are so few BIPOC in conservation are similar to the reasons given for the underrepresentation of BIPOC in ecology in general. More comprehensive perspectives on respectful collaboration with Indigenous communities can be found in Wong et al 2020 (Facets 5:769-783), and in Gewin - Nature 2021 589 pg 315. Perspectives for how to better integrate two-eyed seeing at the graduate level include Massey et al 2021 (Ecology Letters).

Regarding point (b), I have seen fewer articles written specifically on this issue, except perhaps for Chaudhury and Colla (2021), and they come at it from a slightly different angle, although there is overlap between Chaudhury and Colla (2021) and this paper. One query I do have is the use of the phrase Global North. According to Wikipedia, this term includes Asian countries like Japan, Singapore, Taiwan, and South Korea. I think this paper uses Global North to mean the 'white' countries like Australia, New Zealand, USA, Canada, and those in Western Europe. This paper also makes it sound a bit like most of the conservation work around the world is being done by the 'white' countries in the 'Global North'. Is that really true? If so perhaps consider including a figure that shows the fraction of the conservation around the world that is being run by the 'white' Global North. My apologies if I have missed some key points here.

Similar to the point directly above, I felt the paper at times discounted/ignored the great conservation work being done by the Asian Global North, as well as by local organizations in the Global South. E.g. lines 144-146; 229-235. As a personal anecdote, I recently spoke with the head of a Sri Lankan conservation NGO about the lack of racial diversity in 'western' conservation/ecology/evolution. She is Sri Lankan, all of her staff are locals, and they are doing outstanding boots-on-the-ground conservation work on large mammals. According to her there are many people of colour working in conservation in South Asia. How would they fit into the narrative presented in this article? Similarly, one of my colleagues at a Canadian university is an economist of South-Asian descent, from India. He has done some really nice work on human-wildlife conflict in India with Indian academic institutions. Neither of these examples seem to fit into this paper's perspective on who is doing conservation work worldwide.

Personally, I would be more comfortable if the text could be altered slightly to emphasize the key problems associated with the type of conservation that is being done by academic institutions in the 'white' Global North. I don't think the paper is suggesting that all conservation done around the world is flawed, but readers might interpret it that way.

Finally, I don't think that the authors have presented sufficient evidence to support the link that an increase in BIPOC representation in conservation science (in the 'white' Global North) will necessarily break that cycle between the twin spheres. I absolutely agree that diverse voices are needed, but if BIPOC entering this system are being judged and evaluated by the same 'old guard' of academia, how will BIPOC be able to break the cycle? I suppose we need to start somewhere don't we.

Line-by-line comments

39 - Black Lives Matter is not integrated at all in the text. We were not sure why it was included as a keyword/phrase. Also, did you mean to include 'equity' here and not 'equality'?

55- and throughout - both 'equity' and 'equality' are used throughout the paper. Please can the authors double check that they are using the right word in the right context?

69-70- We assume this is written to speak to those who are unaware of these issues, so outlining some examples would be useful.

81- Reference(s) required for the statement "In many places, mainstream conservation has

replaced traditional resource management systems, often to the severe detriment of local people and biodiversity”

92- The paper uses words and phrases such as such as “leaches”, “vicious cycle”, “twin spheres”. These phrases are forceful and convey the importance and urgency of addressing racism and colonialism in conservation. We were a bit conflicted with this phrasing because on one hand the language is powerful and we cheered with fists raised, but on the other hand we worried that the very people who we need to read this paper would not be so enthusiastic. Tip-toeing around white fragility is infuriating but sadly sometimes a necessity.

97- and the surrounding paragraph - “we” is used both to mean the authors, and to mean the field at large. If possible please can you try to be more specific.

99 - echoes Chaudhury and Colla 2020

113- Reference / example would be helpful to the reader for the claim: Contemporary conservation can perpetuate these values, often in spite of strenuous opposition from Indigenous and local people.

114-115 equates colonialism with racism - is this commonly accepted?

116 - What definition of the Global North are you using?

116-117 It's not clear what this sentence means “Recognizing the extent of ecological degradation...” Who is doing the ‘recognizing’? Also, what is ‘true conservation’?

120 - a definition of ‘fortress conservation’ would be helpful here

130 - E.g. Oxford, where many of the co-author currently work; perhaps this paper is a step towards reconciling past wrongs perpetuated by this institution? Many, many institutions still conduct neocolonial conservation work so this is not a dig at Oxford per se but we are mindful that some of the authors are from this institution.

130- examples of the influential NGOs and research institutions would be helpful.

144-169 - these statements have been reviewed in several other comments/papers. It is unclear why they are listed here again; also, there are many outstanding BIPOC conservation practitioners and scholars (but maybe not necessarily working in western universities). I worry a bit that broad statements like 144 discount the contributions made by BIPOC conservation scholars.

1-170 overall this section sometimes felt like a series of statements rather than a narrative that clearly explains how academia and institutional structures work together to produce graduates that have a partial / problematic conception of conservation, and how these structures continue to perpetuate inequalities in who is involved and at what level they are involved in conservation practice and decision making. To me the points made sense, but only because I had already read about these topics.

172-184 - I agree that some degree of underrepresentation of BIPOC in conservation is due to racism. I would argue though, that the references you cite in this section do not necessarily show racism. In contrast, Milkman et al 2015 do show that faculty are more likely to respond to queries from white male students than from all other groups of students, although these findings are not specific to conservation (Journal of Applied Psychology 2015, 100:1678). There's a broader theme that permeates through the paper, and that is that underrepresentation is equated with racism. I agree that in many instances this is true, but some may argue that a lot of underrepresentation comes also from socioeconomic status (eg. aspects of Wanelik et al 2020).

185 - do you know this to be a fact in institutions outside of the ‘white’ Global North?

196-209 - Pettorelli et al 2021 (How international journals can support ecology from the Global South), and Edwards et al 2018 (manuscripts from Asia were 5x more likely to be rejected) could be useful to bolster some of the statements presented here. The Pettorelli paper may be more relevant to conservation.

229-231 - I mentioned this earlier but I'm wondering here about all of the conservation organizations in, for example, Pakistan, India, Philippines, Columbia, etc - are these organizations not doing meaningful work? Or maybe they are just really small compared to some of the large international organizations? This paragraph also seems to be a collection of important sentences that don't necessarily fit well together.

Figures –

1. Please define IPLC

2 a,b These figures are beautifully illustrated. We would like a bit more explanation in each of the figure captions. All three of us struggled a bit with what exactly each part of each figure was depicting. For example, in the first diagram, what are the two people sitting at the table doing? And what is exactly happening on the roof of the 'grants' block?

Decision letter (RSPB-2021-0374.R0)

08-Apr-2021

Dear Ms Rudd:

I am writing to inform you that your manuscript RSPB-2021-0374 entitled "Overcoming racism in the twin spheres of conservation science and practice" has, in its current form, been rejected for publication in Proceedings B.

This action has been taken on the advice of referees, who have recommended that substantial revisions are necessary. With this in mind we would be happy to consider a resubmission, provided the comments of the referees are fully addressed. However please note that this is not a provisional acceptance.

Sincerely,

Dr Maurine Neiman

Associate Editor

Board Member: 1

Comments to Author:

Your manuscript entitled “Overcoming racism in the twin spheres of conservation science and practice” has now undergone careful review by two reviewers, one of whom also had involved some further reviewers. In general, the reviewers appreciated your impassioned writing style, but also expressed some substantial but constructive criticism, much of which is shared across reviews. In particular, the reviewers argued that the manuscript would be more powerful and convincing if the rhetoric is toned down. I agree with this recommendation. The reviewers in particular requested that a revision include fewer over-generalizations, in favor of sticking to the evidence and including more concrete examples. They also recommended consideration of inclusion of issues of intersectionality at certain places in your manuscript. Both reviewers also asked to clarify what exactly is meant by Global South and Global North, to extend upon the examples and experiences in connection with these concepts, and to perhaps even reconsider some of your focus here. Finally, reviewer 1 asked for more clarifications regarding the difference between conservation science and conservation practice. While I am not sure that it would help to replace Figure 1, I agree that the difference could be made even more explicit (though I understand that there is some overlap and that it might be difficult to fully achieve this). I am sure you will also find the further suggestions of the reviewers valuable – please address all of them carefully when revising your paper.

Reviewer(s)' Comments to Author:

Referee: 1

Comments to the Author(s)

This is an impassioned plea for conservation to address issues of systemic racism. The authors focus on both the academic teaching and research environment as well as on the ground practice. The article is generally well-written and reasoned. However, there are a few ways that I think it could be improved.

1. Avoid over-generalizations. While it may seem that the only way to get the conservation community to pay attention is to make bold statements (and as a BIPOC researcher I totally understand the frustration behind the strength of these statements!), your largely non-BIPOC audience will actually dismiss the message if they feel that you are overgeneralizing. To give an example: “Racial stereotypes and often derogatory language used when discussing local communities and field assistants” (line 221) is flavoured by the word “often”. Hang on, you don’t present any survey or statistics to demonstrate that most conservation scientists use derogatory language most of the time when discussing local communities and assistants. Instead, I suspect that this statement is based on your personal experience, which of course is valid but nonetheless limited (too limited to use the word “often”). You could say, “In our experience, when racial stereotypes and derogatory language are used” or “Racial stereotypes and derogatory language are too often used”. This is one example, but I felt this issue permeated the manuscript – I suggest tightening up the writing, carefully explaining what is the lived experience of the authors, and what are the conclusions of studies, and keeping the language used consistent with the source of the information would actually give this more punch in the end.

2. Stick tight to the evidence. For example, the authors write (line 159) “Hobbies such as birding and hiking require costly equipment [28].” While binoculars are definitely a cost barrier, I was intrigued what the costly hiking equipment might be, given that day hikes can be done in runners – so looked up the original article. The original article is actually about field courses, involving weeks of 8-10 hours a day in the field, and requiring sturdy hiking boots, backpacks, rain gear, etc. Again, if your audience thinks that you are playing loose with the facts, you will lose credibility. (as a sidebar, I think travel to hiking destinations and concerns about field safety is probably more of a barrier for casual hiking than footwear).

3. Provide concrete examples. While there are a number of useful examples presented here and there in the manuscript, I would have liked the ratio of examples to generalizations to be much higher. I do appreciate that you provide references for many of the broad statements you make, but the causal reader will not be familiar with all of these. Instead, providing a concrete example or statistic would be much more persuasive. For example, "These practices came at great cost to local people, including through forced removal, abuse, and murder [11]." (line 109). Murder! That may seem sensational to your audience. But if you used a sentence to outline an example from this reference that involved murder, then you'll be much more convincing.

4. Clarify relationship between BIPOC in Global North and Global South, and clearly differentiate the differences in their experiences of racism. Right now, there is little distinguishing between the unique experience of these two populations. And in fact, I felt that in some sections (like Exclusion from Engaging with Nature), the concerns of BIPOC in the North overtook the South. BIPOC in the North are dealing with being minorities in a Eurocentric and majority white culture. However, this doesn't mean that this experience makes us (I'll include myself) models of anti-oppression when it comes to engaging BIPOC in the South in conservation. Meanwhile BIPOC in the South have very different challenges – even when in a majority BIPOC country, there are intersectional issues (e.g. gender oppression, sexual orientation, access to economic privilege) that affect them both in terms of local engagement with the conservation community as well as attempting to gain academic credentials.

5. Restructure manuscript to clarify the "conservation science" vs. "conservation practice" division. In terms of the headings of sections, I was sometimes confused which we were talking about. I also did not find Figure 1 that useful: the points were vague, and it is never clear how conservation practice affects conservation science. You might instead consider a point form action plan for the conservation community that addresses the points you raise in text. Figure 2 was nicely done and engaging.

Line 48 – I believe it should be "syllabi" not "syllabuses"

Line 78 – define "traditional resource management systems". Are you referring exclusively to Indigenous resource management systems, and if not, what is the time frame and context for your definition?

Line 93. "preconceptions that conservationist graduates" instead of "preconceptions conservationist graduates"

Line 95. You describe "This vicious cycle" but above only describe conservation science -> conservation practice. For a cycle, you also need the arrow to go the other way.

Line 103. "people that conservation has harmed and continues to harm," instead of "people conservation has and continues to harm,"

Line 231 "hoFmbizping" ...?

Figure 1: please do not use acronym (IPLC) in figure with no explanation in legend.

Referee: 2

Comments to the Author(s)

Review of Overcoming racism in the twin spheres of conservation science and practice
RSPB-2021-0374

We thank you for the opportunity to review this Letter. For context, three of us read the paper and we compiled our comments into one review. Our research expertise is broadly in ecology, evolution, and conservation. The three of us are also very engaged in equity, diversity, and inclusion work, although we are not scholars of critical race theory. We are a mix of males and females, BIPOC and non-BIPOC. We use I/We a bit loosely.

First and foremost, thank you for the time, effort, and emotional energy it took to write this piece. There is much injustice in the world but also there are many who have benefitted from the status quo and are reluctant to change. Although many of these types of commentaries have been

written lately, there should be no limits on the number of times we say that change needs to happen. Having said that, our goal here is not to discount any of the authors' words or lived experiences, rather our goal is to try to provide feedback that strengthens what we feel are the core points of the paper.

General comments

The main points of this letter seem to be:

- (a) Black, Indigenous, and people of colour are underrepresented in conservation biology in the academy;
- (b) conventional western-led conservation practice is rooted in colonialism and racism; modern-day approaches to conservation (e.g. parachute science) ignore and disadvantage Indigenous voices and local communities (largely from the Global South).
- (c) Underrepresentation of BIPOC in conservation science perpetuates neocolonial and racist conservation practices, and vice versa.

Regarding point (a), several recent articles have highlighted that BIPOC are underrepresented in the academy, and especially in ecology-related fields. Some articles have focused on Academia in general (Barber et al 2020) while others have focused on ecology, evolution, and conservation (Chaudhury and Colla 2020, Massey et al 2021, O'Brien et al 2020, Wanelik et al 2020, Graves 2019). Because so many excellent papers have also been written on this topic, we are unclear as to how this current paper adds to the existing discourse. The reasons for why there are so few BIPOC in conservation are similar to the reasons given for the underrepresentation of BIPOC in ecology in general. More comprehensive perspectives on respectful collaboration with Indigenous communities can be found in Wong et al 2020 (Facets 5:769-783), and in Gewin - Nature 2021 589 pg 315. Perspectives for how to better integrate two-eyed seeing at the graduate level include Massey et al 2021 (Ecology Letters).

Regarding point (b), I have seen fewer articles written specifically on this issue, except perhaps for Chaudhury and Colla (2021), and they come at it from a slightly different angle, although there is overlap between Chaudhury and Colla (2021) and this paper. One query I do have is the use of the phrase Global North. According to Wikipedia, this term includes Asian countries like Japan, Singapore, Taiwan, and South Korea. I think this paper uses Global North to mean the 'white' countries like Australia, New Zealand, USA, Canada, and those in Western Europe. This paper also makes it sound a bit like most of the conservation work around the world is being done by the 'white' countries in the 'Global North'. Is that really true? If so perhaps consider including a figure that shows the fraction of the conservation around the world that is being run by the 'white' Global North. My apologies if I have missed some key points here.

Similar to the point directly above, I felt the paper at times discounted/ignored the great conservation work being done by the Asian Global North, as well as by local organizations in the Global South. E.g. lines 144-146; 229-235. As a personal anecdote, I recently spoke with the head of a Sri Lankan conservation NGO about the lack of racial diversity in 'western' conservation/ecology/evolution. She is Sri Lankan, all of her staff are locals, and they are doing outstanding boots-on-the-ground conservation work on large mammals. According to her there are many people of colour working in conservation in South Asia. How would they fit into the narrative presented in this article? Similarly, one of my colleagues at a Canadian university is an economist of South-Asian descent, from India. He has done some really nice work on human-wildlife conflict in India with Indian academic institutions. Neither of these examples seem to fit into this paper's perspective on who is doing conservation work worldwide.

Personally, I would be more comfortable if the text could be altered slightly to emphasize the key problems associated with the type of conservation that is being done by academic institutions in the 'white' Global North. I don't think the paper is suggesting that all conservation done around the world is flawed, but readers might interpret it that way.

Finally, I don't think that the authors have presented sufficient evidence to support the link that an increase in BIPOC representation in conservation science (in the 'white' Global North) will necessarily break that cycle between the twin spheres. I absolutely agree that diverse voices are needed, but if BIPOC entering this system are being judged and evaluated by the same 'old guard' of academia, how will BIPOC be able to break the cycle? I suppose we need to start somewhere don't we.

Line-by-line comments

39 – Black Lives Matter is not integrated at all in the text. We were not sure why it was included as a keyword/phrase. Also, did you mean to include 'equity' here and not 'equality'?

55- and throughout – both 'equity' and 'equality' are used throughout the paper. Please can the authors double check that they are using the right word in the right context?

69-70- We assume this is written to speak to those who are unaware of these issues, so outlining some examples would be useful.

81- Reference(s) required for the statement "In many places, mainstream conservation has replaced traditional resource management systems, often to the severe detriment of local people and biodiversity"

92- The paper uses words and phrases such as such as "leaches", "vicious cycle", "twin spheres". These phrases are forceful and convey the importance and urgency of addressing racism and colonialism in conservation. We were a bit conflicted with this phrasing because on one hand the language is powerful and we cheered with fists raised, but on the other hand we worried that the very people who we need to read this paper would not be so enthusiastic. Tip-toeing around white fragility is infuriating but sadly sometimes a necessity.

97- and the surrounding paragraph - "we" is used both to mean the authors, and to mean the field at large. If possible please can you try to be more specific.

99 – echoes Chaudhury and Colla 2020

113- Reference / example would be helpful to the reader for the claim: Contemporary conservation can perpetuate these values, often in spite of strenuous opposition from Indigenous and local people.

114-115 equates colonialism with racism – is this commonly accepted?

116 - What definition of the Global North are you using?

116-117 It's not clear what this sentence means "Recognizing the extent of ecological degradation..." Who is doing the 'recognizing'? Also, what is 'true conservation'?

120 – a definition of 'fortress conservation' would be helpful here

130 – E.g. Oxford, where many of the co-author currently work; perhaps this paper is a step towards reconciling past wrongs perpetuated by this institution? Many, many institutions still conduct neocolonial conservation work so this is not a dig at Oxford per se but we are mindful that some of the authors are from this institution.

130- examples of the influential NGOs and research institutions would be helpful.

144-169 – these statements have been reviewed in several other comments/papers. It is unclear why they are listed here again; also, there are many outstanding BIPOC conservation practitioners and scholars (but maybe not necessarily working in western universities). I worry a bit that broad statements like 144 discount the contributions made by BIPOC conservation scholars.

1-170 overall this section sometimes felt like a series of statements rather than a narrative that clearly explains how academia and institutional structures work together to produce graduates that have a partial / problematic conception of conservation, and how these structures continue to perpetuate inequalities in who is involved and at what level they are involved in conservation practice and decision making. To me the points made sense, but only because I had already read about these topics.

172-184 – I agree that some degree of underrepresentation of BIPOC in conservation is due to racism. I would argue though, that the references you cite in this section do not necessarily show racism. In contrast, Milkman et al 2015 do show that faculty are more likely to respond to queries from white male students than from all other groups of students, although these findings are not

specific to conservation (Journal of Applied Psychology 2015, 100:1678). There's a broader theme that permeates through the paper, and that is that underrepresentation is equated with racism. I agree that in many instances this is true, but some may argue that a lot of underrepresentation comes also from socioeconomic status (eg. aspects of Wanelik et al 2020).

185 – do you know this to be a fact in institutions outside of the 'white' Global North?

196-209 – Pettorelli et al 2021 (How international journals can support ecology from the Global South), and Edwards et al 2018 (manuscripts from Asia were 5x more likely to be rejected) could be useful to bolster some of the statements presented here. The Pettorelli paper may be more relevant to conservation.

229-231 – I mentioned this earlier but I'm wondering here about all of the conservation organizations in, for example, Pakistan, India, Philippines, Columbia, etc – are these organizations not doing meaningful work? Or maybe they are just really small compared to some of the large international organizations? This paragraph also seems to be a collection of important sentences that don't necessarily fit well together.

Figures –

1. Please define IPLC

2 a,b These figures are beautifully illustrated. We would like a bit more explanation in each of the figure captions. All three of us struggled a bit with what exactly each part of each figure was depicting. For example, in the first diagram, what are the two people sitting at the table doing? And what is exactly happening on the roof of the 'grants' block?

Author's Response to Decision Letter for (RSPB-2021-0374.R0)

See Appendix A.

RSPB-2021-1871.R0

Review form: Reviewer 1

Recommendation

Accept with minor revision (please list in comments)

Scientific importance: Is the manuscript an original and important contribution to its field?

Acceptable

General interest: Is the paper of sufficient general interest?

Good

Quality of the paper: Is the overall quality of the paper suitable?

Good

Is the length of the paper justified?

Yes

Should the paper be seen by a specialist statistical reviewer?

No

Do you have any concerns about statistical analyses in this paper? If so, please specify them explicitly in your report.

No

It is a condition of publication that authors make their supporting data, code and materials available - either as supplementary material or hosted in an external repository. Please rate, if applicable, the supporting data on the following criteria.

Is it accessible?

N/A

Is it clear?

N/A

Is it adequate?

N/A

Do you have any ethical concerns with this paper?

No

Comments to the Author

Thank you for this revised manuscript, and the evident care that you took with the revisions. Figure 1 is much better integrated with the manuscript and I appreciate that statements are better supported with examples and references. I just have a few extra comments.

Line 85 “in countries which are BIPOC majority and BIPOC minority.” -> regardless of whether the country is BIPOC majority and BIPOC minority.

Line 101 extra space in “programmes , which”

Line 158 “Influential researchers and advocacy groups based in the Global North stridently advocate for extending their conservation ideologies to vastly different socioecological and cultural contexts, with seemingly no regard for traditional practices or ethics in those locations [25]. These reckless prescriptions” ...tone it down here, words like “stridently, seemingly, reckless” do not impart extra information to the sentence, are more rhetorical.

Line 182 has an abrupt shift from representation exclusion of BIPOC people from conservation decision making to physical exclusion -and then financial exclusion- of BIPOC people from outdoor space. I think you need to make a smoother transition between these two types of exclusion (different, although stemming from same racist paradigms).

Line 256 extra period in “relate. . Racial”

Line 270 I found “almost ‘Brahminical’ reverence” jarring, especially in the context of this essay. Do we really have to appropriate a Hindu term here? It’s as bad as people talking of scared cows. I’m not saying this because I support the hierarchy of the Hindu caste system but because I think we need to not cavalierly borrow references from BIPOC-dominated religions.

Review form: Reviewer 2

Recommendation

Reject – article is not of sufficient interest (we will consider a transfer to another journal)

Scientific importance: Is the manuscript an original and important contribution to its field?

Marginal

General interest: Is the paper of sufficient general interest?

Marginal

Quality of the paper: Is the overall quality of the paper suitable?

Marginal

Is the length of the paper justified?

Yes

Should the paper be seen by a specialist statistical reviewer?

No

Do you have any concerns about statistical analyses in this paper? If so, please specify them explicitly in your report.

No

It is a condition of publication that authors make their supporting data, code and materials available - either as supplementary material or hosted in an external repository. Please rate, if applicable, the supporting data on the following criteria.

Is it accessible?

N/A

Is it clear?

N/A

Is it adequate?

N/A

Do you have any ethical concerns with this paper?

No

Comments to the Author

Review of RSPB-2021-1872 Overcoming racism in the twin spheres of conservation science and practice.

This is my second time reviewing this manuscript. The authors sort of addressed my concerns in the first round, mostly by way of addressing reviewer 1's concerns.

My main concerns from round one included that I did not really see how this piece differed from similar articles that have been published in the last two years. Haelewaters et al 2021 (Ten simple rules for Global North researchers to stop perpetuating helicopter research in the Global South) has also recently been published in PLoS Computational Biology, and it offers concrete steps for north-south collaborations. Cronin et al 2021 (Anti-racist interventions to transform ecology, evolution and conservation biology departments) has also been published recently and the authors do cite this paper.

The main messages that I get from this manuscript are a) conservation has a deep-seated racist past; b) BIPOC people are excluded from conservation science and practice.

In the response to reviewers, the authors write that "This paper adds further to the existing discourse by focusing specifically on both conservation science and practice. We identify

problems that are unique to each of these spheres and the way in which they perpetuate one another, laid out against the context of conservation history.”

I’ve read the revised ms over a few times now and I am still having trouble linking problems in conservation practice to problems in conservation science. (Evidence for the conservation science perpetuating problems in conservation practice is nicely discussed in the paper.) Maybe this information is in lines 112-113? Along these lines, the manuscript both talks about ‘largely independent spheres’ as well as ‘vicious cycle’. Are the cycles strictly within the spheres or is there one cycle that goes from one sphere to the next... and the latter, how can the spheres be largely independent.

The majority of the paper reviews the different ways that BIPOC can be excluded from conservation-related courses or from accessing outdoor spaces, and illustrates instances where BIPOC have been mistreated in academia or conservation practice. These statements have been published elsewhere and there is not much that is novel here. I’m uncomfortable with the level of generalizations regarding BIPOC throughout the pages, and I’ll push back a bit and say that there are many cultures where it’s just not the cultural norm to go hiking – it’s not that folks feel unwelcome, it’s just not something that’s done regularly.

As someone who is not a white Caucasian, personally I’m a bit tired of all of publications outlining mistreatment against BIPOC. I would have liked the paper more to focus on concrete solutions and less on repeating what’s already been said. Maybe this paper will resonate more with non-BIPOC scholars. Solutions are presented in boxes 1-3 but they are fairly sweeping and high-level statements and not as helpful as some of the other solutions that have already been presented in the literature (e.g. Haelewaters et al 2021, or Cronin et al 2021, others). Also, decolonizing the curriculum is not the same as increasing BIPOC content into the curriculum.

I also have a really hard time with the overall sentiment of this paper that racism in conservation science and practice can be fixed by making the fields more inclusive to BIPOC scholars and practitioners. I know plenty of racist, culturally illiterate BIPOC folks.

Given all of the papers published on similar topics in this area, I would think about what can you do with this paper that would affect the most change. I think publishing it as a regular paper in a journal will limit its utility. Who do you want to read this paper and why, and how do you get into in their hands.

Decision letter (RSPB-2021-1871.R0)

23-Sep-2021

Dear Ms Rudd

I am pleased to inform you that your manuscript RSPB-2021-1871 entitled "Overcoming racism in the twin spheres of conservation science and practice" has been accepted for publication, pending some revisions, in Proceedings B. Indeed, I consider the appropriate decision to be somewhere between "Accept with minor revision" and "Revise". I ultimately chose the former to avoid the automatic indication that the paper will be sent out again for review, and I do not think that additional review will be necessary. Nevertheless, I do think that you need to attend carefully to the critiques regarding novelty, nuance, the addition of concrete and actionable suggestions, and the need for more connections between problems and practice laid out by reviewer 2, along with making the changes suggested by reviewer 1.

Therefore, I invite you to respond to the referee(s)' comments and revise your manuscript within 14 days. If you do not think you will be able to meet this date please let us know.

[http://datadryad.org/submit?journalID=RSPB&manu=\(Document not available\)](http://datadryad.org/submit?journalID=RSPB&manu=(Document%20not%20available)) which will take you to your unique entry in the Dryad repository. If you have already submitted your data to dryad you can make any necessary revisions to your dataset by following the above link. Please see <https://royalsociety.org/journals/ethics-policies/data-sharing-mining/> for more details.

Sincerely,

Dr Maurine Neiman

Associate Editor

Board Member

Comments to Author:

I have now received two reviews from the original two referees. Reviewer one was very happy with the revisions, and only asks for some minor changes and edits. Reviewer 2 now sees less value in the paper, mainly from the perspective that the manuscript no longer reflects new thinking in the field. I did take this recommendation seriously. However, as associate editor, I believe that a scientifically sound paper still has substantial value even if other papers have come out on the topic in the meantime. This is why my recommendation is to focus on implementation of the changes asked for by reviewer 1. That said, please also address a bit more in detail how problems in conservation practice link to problems in conservation science, as asked for by reviewer 2.

Reviewer(s)' Comments to Author:

Referee: 1

Comments to the Author(s).

Thank you for this revised manuscript, and the evident care that you took with the revisions.

Figure 1 is much better integrated with the manuscript and I appreciate that statements are better supported with examples and references. I just have a few extra comments.

Line 85 “in countries which are BIPOC majority and BIPOC minority.” –> regardless of whether the country is BIPOC majority and BIPOC minority.

Line 101 extra space in “programmes , which”

Line 158 “Influential researchers and advocacy groups based in the Global North stridently advocate for extending their conservation ideologies to vastly different socioecological and cultural contexts, with seemingly no regard for traditional practices or ethics in those locations [25]. These reckless prescriptions” ...tone it down here, words like “stridently, seemingly, reckless” do not impart extra information to the sentence, are more rhetorical.

Line 182 has an abrupt shift from representation exclusion of BIPOC people from conservation decision making to physical exclusion -and then financial exclusion- of BIPOC people from

outdoor space. I think you need to make a smoother transition between these two types of exclusion (different, although stemming from same racist paradigms).

Line 256 extra period in “relate. . Racial”

Line 270 I found “almost ‘Brahminical’ reverence” jarring, especially in the context of this essay. Do we really have to appropriate a Hindu term here? It’s as bad as people talking of scared cows. I’m not saying this because I support the hierarchy of the Hindu caste system but because I think we need to not cavalierly borrow references from BIPOC-dominated religions.

Referee: 2

Comments to the Author(s).

Review of RSPB-2021-1872 Overcoming racism in the twin spheres of conservation science and practice.

This is my second time reviewing this manuscript. The authors sort of addressed my concerns in the first round, mostly by way of addressing reviewer 1’s concerns.

My main concerns from round one included that I did not really see how this piece differed from similar articles that have been published in the last two years. Haelewaters et al 2021 (Ten simple rules for Global North researchers to stop perpetuating helicopter research in the Global South) has also recently been published in PLoS Computational Biology, and it offers concrete steps for north-south collaborations. Cronin et al 2021 (Anti-racist interventions to transform ecology, evolution and conservation biology departments) has also been published recently and the authors do cite this paper.

The main messages that I get from this manuscript are a) conservation has a deep-seated racist past; b) BIPOC people are excluded from conservation science and practice.

In the response to reviewers, the authors write that “This paper adds further to the existing discourse by focusing specifically on both conservation science and practice. We identify problems that are unique to each of these spheres and the way in which they perpetuate one another, laid out against the context of conservations history.”

I’ve read the revised ms over a few times now and I am still having trouble linking problems in conservation practice to problems in conservation science. (Evidence for the conservation science perpetuating problems in conservation practice is nicely discussed in the paper.) Maybe this information is in lines 112-113? Along these lines, the manuscript both talks about ‘largely independent spheres’ as well as ‘vicious cycle’. Are the cycles strictly within the spheres or is there one cycle that goes from one sphere to the next... and the latter, how can the spheres be largely independent.

The majority of the paper reviews the different ways that BIPOC can be excluded from conservation-related courses or from accessing outdoor spaces, and illustrates instances where BIPOC have been mistreated in academia or conservation practice. These statements have been published elsewhere and there is not much that is novel here. I’m uncomfortable with the level of generalizations regarding BIPOC throughout the pages, and I’ll push back a bit and say that there are many cultures where it’s just not the cultural norm to go hiking – it’s not that folks feel unwelcome, it’s just not something that’s done regularly.

As someone who is not a white Caucasian, personally I’m a bit tired of all of publications outlining mistreatment against BIPOC. I would have liked the paper more to focus on concrete solutions and less on repeating what’s already been said. Maybe this paper will resonate more with non-BIPOC scholars. Solutions are presented in boxes 1-3 but they are fairly sweeping and high-level statements and not as helpful as some of the other solutions that have already been

presented in the literature (e.g. Haelewaters et al 2021, or Cronin et al 2021, others). Also, decolonizing the curriculum is not the same as increasing BIPOC content into the curriculum.

I also have a really hard time with the overall sentiment of this paper that racism in conservation science and practice can be fixed by making the fields more inclusive to BIPOC scholars and practitioners. I know plenty of racist, culturally illiterate BIPOC folks.

Given all of the papers published on similar topics in this area, I would think about what can you do with this paper that would affect the most change. I think publishing it as a regular paper in a journal will limit its utility. Who do you want to read this paper and why, and how do you get into in their hands.

Author's Response to Decision Letter for (RSPB-2021-1871.R0)

See Appendix B.

Decision letter (RSPB-2021-1871.R1)

07-Oct-2021

Dear Ms Rudd

I am pleased to inform you that your manuscript entitled "Overcoming racism in the twin spheres of conservation science and practice" has been accepted for publication in Proceedings B.

Data Accessibility section

Open Access

Paper charges

Sincerely,
Editor, Proceedings B
mailto: proceedingsb@royalsociety.org

Appendix A

Rudd et al. Overcoming racism in the twin spheres of conservation

Response to review

Associate Editor

AE1. Your manuscript entitled “Overcoming racism in the twin spheres of conservation science and practice” has now undergone careful review by two reviewers, one of whom also had involved some further reviewers. In general, the reviewers appreciated your impassioned writing style, but also expressed some substantial but constructive criticism, much of which is shared across reviews. In particular, the reviewers argued that the manuscript would be more powerful and convincing if the rhetoric is toned down. I agree with this recommendation.

We gratefully received the comments made by the reviewers and have striven to address them all. We have carefully considered the tone of language used throughout and edited this where appropriate. In addition, we have added several citations to bolster claims which were identified as being too strong in the absence of supporting information.

AE2. The reviewers in particular requested that a revision include fewer over-generalizations, in favor of sticking to the evidence and including more concrete examples. They also recommended consideration of inclusion of issues of intersectionality at certain places in your manuscript.

We have clearly stated where we are speaking from personal experience of the authors. Where we are not, we have included appropriate citations and/or examples. We have considered intersectionality at various points in the manuscript, but we believe that deeper comment on this would require a great expansion of the manuscript in terms of both concept and content. While we are all in agreement that this is of great importance, we do not believe this manuscript is the appropriate place for such an in-depth analysis.

AE3. Both reviewers also asked to clarify what exactly is meant by Global South and Global North, to extend upon the examples and experiences in connection with these concepts, and to perhaps even reconsider some of your focus here.

We were very grateful for these comments and have thought carefully about the best terms throughout the manuscript, which often differ depending on the context of the sentence/paragraph. We have made amendments throughout to address this, and we have also added a short paragraph to the introduction, acknowledging the variety and breadth of experiences that BIPOC people in conservation will have depending on social and economic factors (beginning on line 80). We have also expanded our commentary on the experiences of individuals outside of the Global North in response to reviewers’ comments (please see our response to R1.5 below).

AE4. Finally, reviewer 1 asked for more clarifications regarding the difference between conservation science and conservation practice.

We have made a series of edits to the introduction (beginning on line 112), and we have significantly reworked Figure 1 to convey more clearly the differences between

conservation science and conservation practice, and the cycles of reinforcement between them.

AE5. While I am not sure that it would help to replace Figure 1, I agree that the difference could be made even more explicit (though I understand that there is some overlap and that it might be difficult to fully achieve this).

We have reworked Figure 1 to help clarify the distinction between conservation science and practice, as well as the cycle of reinforcement between these two spheres.

AE6. I am sure you will also find the further suggestions of the reviewers valuable – please address all of them carefully when revising your paper.

We are grateful for the reviewers' thorough and thoughtful comments, which mean this revision improves substantially on our initial submission.

Referee: 1

R1.1. This is an impassioned plea for conservation to address issues of systemic racism. The authors focus on both the academic teaching and research environment as well as on the ground practice. The article is generally well-written and reasoned. However, there are a few ways that I think it could be improved.

We thank reviewer 1 for their constructive comments and suggestions, which helped us make several improvements in this revision.

R1.2. Avoid over-generalizations. While it may seem that the only way to get the conservation community to pay attention is to make bold statements (and as a BIPOC researcher I totally understand the frustration behind the strength of these statements!), your largely non-BIPOC audience will actually dismiss the message if they feel that you are overgeneralizing. To give an example: “Racial stereotypes and often derogatory language used when discussing local communities and field assistants” (line 221) is flavoured by the word “often”. Hang on, you don't present any survey or statistics to demonstrate that most conservation scientists use derogatory language most of the time when discussing local communities and assistants. Instead, I suspect that this statement is based on your personal experience, which of course is valid but nonetheless limited (too limited to use the word “often”). You could say, “In our experience, when racial stereotypes and derogatory language are used” or “Racial stereotypes and derogatory language are too often used”. This is one example, but I felt this issue permeated the manuscript – I suggest tightening up the writing, carefully explaining what is the lived experience of the authors, and what are the conclusions of studies, and keeping the language used consistent with the source of the information would actually give this more punch in the end.

We agree fully with reviewer 1's comments here and have corrected the overgeneralisations (including the specific example given here on line 457) throughout the manuscript. In some places, we have specifically referred to our own experiences and so have explained that. In others, we have included additional or more appropriate

citations to back up the claims we are making.

R1.3. Stick tight to the evidence. For example, the authors write (line 159) “Hobbies such as birding and hiking require costly equipment [28].” While binoculars are definitely a cost barrier, I was intrigued what the costly hiking equipment might be, given that day hikes can be done in runners – so looked up the original article. The original article is actually about field courses, involving weeks of 8-10 hours a day in the field, and requiring sturdy hiking boots, backpacks, rain gear, etc. Again, if your audience thinks that you are playing loose with the facts, you will lose credibility. (as a sidebar, I think travel to hiking destinations and concerns about field safety is probably more of a barrier for casual hiking than footwear).

We thank you for bringing this to our attention, and we have corrected the citation in this example (see line 280). We have also taken care to go through all the citations in the manuscript and ensure we are quoting the facts accurately as presented in the original article. This has led to us removing some citations, replacing them with better citations or with statements that we are drawing on personal experiences.

R1.4. Provide concrete examples. While there are a number of useful examples presented here and there in the manuscript, I would have liked the ratio of examples to generalizations to be much higher. I do appreciate that you provide references for many of the broad statements you make, but the casual reader will not be familiar with all of these. Instead, providing a concrete example or statistic would be much more persuasive. For example, “These practices came at great cost to local people, including through forced removal, abuse, and murder [11].” (line 109). Murder! That may seem sensational to your audience. But if you used a sentence to outline an example from this reference that involved murder, then you’ll be much more convincing.

We appreciate this comment and agree that using in text examples does bolster statements and arguments. We include some specific examples (see lines 207, 246, 263) which believe are particularly impactful. However, there are multiple constraints that restrict us from adding more. Firstly, the amount of empirical research on these topics is severely lacking, so finding published statistics that are widely applicable (and not solely focused on the USA or UK) is difficult. We do not want to over emphasise examples from these regions as it will detract from the geographic extent of the arguments we present. Further we are constrained by word count when deciding how many examples to elaborate on in text.

R1.5. Clarify relationship between BIPOC in Global North and Global South, and clearly differentiate the differences in their experiences of racism. Right now, there is little distinguishing between the unique experience of these two populations. And in fact, I felt that in some sections (like Exclusion from Engaging with Nature), the concerns of BIPOC in the North overtook the South. BIPOC in the North are dealing with being minorities in a Eurocentric and majority white culture. However, this doesn’t mean that this experience makes us (I’ll include myself) models of anti-oppression when it comes to engaging BIPOC in the South in conservation. Meanwhile BIPOC in the South have very different challenges – even when in a majority BIPOC country, there are intersectional issues (e.g. gender

oppression, sexual orientation, access to economic privilege) that affect them both in terms of local engagement with the conservation community as well as attempting to gain academic credentials.

We thank reviewer 1 for these comments, which caused us to think deeply about the message we are trying to convey. As per our response to comment AE3, we have added a paragraph to the introduction to directly address our view on the similarities and differences in experiences of racism both between the Global North and Global South, and amongst individuals within these places. We agree with reviewer 1 that there are unique experiences between these populations, but we also believe there is overlap due to the international nature of the conservation community. Therefore, we have chosen to draw attention to the fact that overwhelmingly, BIPOC people are minorities in the “conservation space” regardless of where they come from. This is because the Eurocentric, white culture you refer to as being inherent to the Global North, is inherent to conservation broadly. Further, we believe the intersectional issues you raise apply to individuals in both the Global North and Global South (although of course to varying degrees, depending on a suite of other factors). In response to this comment, multiple authors made revisions to the manuscript to address the disparity in emphasis on issues in the Global North and Global South, particularly in the section “Exclusion from Engaging with Nature” (see paragraph beginning line 278).

R1.6. Restructure manuscript to clarify the “conservation science” vs. “conservation practice” division. In terms of the headings of sections, I was sometimes confused which we were talking about.

We made edits to the subheadings to specifically include the terms “conservation science” and “conservation practice” and were also careful to state in text whether we were referring to one, the other, or both.

R1.7. I also did not find Figure 1 that useful: the points were vague, and it is never clear how conservation practice affects conservation science. You might instead consider a point form action plan for the conservation community that addresses the points you raise in text. Figure 2 was nicely done and engaging.

We are very grateful to R1 for making us think carefully about the role of Figure 1. We have substantially reworked the figure, its legend, and how we introduce it in the manuscript text, and we believe it now does more (and more useful) work in helping us make our argument. We state that overcoming the vicious cycle we depict in Figure 1 will require changes to individual and collective behaviours, described in the text boxes- so chose not the replace Figure 1 with a point form action plan.

R1.8. Line 48 – I believe it should be “syllabi” not “syllabuses”

We believe both “syllabi” and “syllabuses” are correct, so we have left this as it was for the time being. If our manuscript is eventually accepted, we will use whichever version matches the journal’s style.

R1.9. Line 78 – define “traditional resource management systems”. Are you referring exclusively to Indigenous resource management systems, and if not, what is the time frame and context for your definition?

We have now replaced this term with the more commonly used “IPLCs”, which we have been sure to define in text.

R1.10. Line 93. “preconceptions that conservationist graduates” instead of “preconceptions conservationist graduates”

We have edited as suggested, with thanks.

R1.11. Line 95. You describe “This vicious cycle” but above only describe conservation science -> conservation practice. For a cycle, you also need the arrow to go the other way.

We have made several changes to this section of the manuscript to make clearer the bidirectional nature of the vicious cycle. Please see response to R1.6 for further details.

R1.12.Line 103. “people that conservation has harmed and continues to harm,” instead of “people conservation has and continues to harm,”

Appropriate edits have been made to the text here.

R1.13. Line 231 “hoFmbizping”...?

This has been corrected.

R1.14. Figure 1: please do not use acronym (IPLC) in figure with no explanation in legend.

Thank you for drawing our attention to this oversight, which has now been rectified.

Referee: 2

R2.1. We thank you for the opportunity to review this Letter. For context, three of us read the paper and we compiled our comments into one review. Our research expertise is broadly in ecology, evolution, and conservation. The three of us are also very engaged in equity, diversity, and inclusion work, although we are not scholars of critical race theory. We are a mix of males and females, BIPOC and non-BIPOC. We use I/We a bit loosely.

First and foremost, thank you for the time, effort, and emotional energy it took to write this piece. There is much injustice in the world but also there are many who have benefitted from

the status quo and are reluctant to change. Although many of these types of commentaries have been written lately, there should be no limits on the number of times we say that change needs to happen. Having said that, our goal here is not to discount any of the authors' words or lived experiences, rather our goal is to try to provide feedback that strengthens what we feel are the core points of the paper.

General comments

The main points of this letter seem to be:

- (a) Black, Indigenous, and people of colour are underrepresented in conservation biology in the academy;
- (b) conventional western-led conservation practice is rooted in colonialism and racism; modern-day approaches to conservation (e.g. parachute science) ignore and disadvantage Indigenous voices and local communities (largely from the Global South).
- (c) Underrepresentation of BIPOC in conservation science perpetuates neocolonial and racist conservation practices, and vice versa.

We are very grateful for all of the reviewers' comments and for expressing their support of manuscripts such as this. We are especially grateful for the succinct summary of a key component of our argument in point (c), and have used it to help clarify the relationships between conservation science and conservation practice.

R2.2. Regarding point (a), several recent articles have highlighted that BIPOC are underrepresented in the academy, and especially in ecology-related fields. Some articles have focused on Academia in general (Barber et al 2020) while others have focused on ecology, evolution, and conservation (Chaudhury and Colla 2020, Massey et al 2021, O'Brien et al 2020, Wanelik et al 2020, Graves 2019). Because so many excellent papers have also been written on this topic, we are unclear as to how this current paper adds to the existing discourse.

We agree with the reviewers that many excellent papers have recently been published on related topics, many of which we have cited in this manuscript. This paper adds further to the existing discourse by focusing specifically on both conservation science and practice. We identify problems that are unique to each of these spheres and the way in which they perpetuate one another, laid out against the context of conservation history. We also provide recommendations on how best to break these reinforcing cycles, coming from the multiple perspectives of our vastly diverse authorship. To our knowledge, such content has not been covered in a published paper to date.

R2.3. The reasons for why there are so few BIPOC in conservation are similar to the reasons given for the underrepresentation of BIPOC in ecology in general.

We agree that there are overlaps, but there are also unique factors influencing the conservation field. We have not compared the factors leading to underrepresentation of BIPOC in ecology vs conservation as that is not the purpose of this manuscript, but we do believe that it is important for that content to be covered here to justify the recommendations made.

R2.4. More comprehensive perspectives on respectful collaboration with Indigenous communities can be found in Wong et al 2020 (Facets 5:769-783), and in Gewin - Nature 2021 589 pg 315. Perspectives for how to better integrate two-eyed seeing at the graduate level include Massey et al 2021 (Ecology Letters).

We agree that these papers are excellent examples of the work that has recently been published on collaboration with Indigenous communities and the need for equity for BIPOC scholars in ecology and evolutionary biology, and they have been cited in the manuscript. Please see our response to R2.2 and R2.3 for further comment.

R2.5. Regarding point (b), I have seen fewer articles written specifically on this issue, except perhaps for Chaudhury and Colla (2021), and they come at it from a slightly different angle, although there is overlap between Chaudhury and Colla (2021) and this paper.

We agree that there is some overlap with the Chaudhury and Colla (2021) paper which we cite in this manuscript, but overall, we think that our approach and content is different but complementary, as outlined in our responses to R2.2 and R2.3

R2.6. One query I do have is the use of the phrase Global North. According to Wikipedia, this term includes Asian countries like Japan, Singapore, Taiwan, and South Korea. I think this paper uses Global North to mean the 'white' countries like Australia, New Zealand, USA, Canada, and those in Western Europe.

We are grateful for this comment and have taken care to edit the manuscript and ensure we are using the best terms throughout. In some places we do believe that using Global North/South is accurate, but in others we have opted to use BIPOC majority/minority.

R2.7. This paper also makes it sound a bit like most of the conservation work around the world is being done by the 'white' countries in the 'Global North'. Is that really true? If so perhaps consider including a figure that shows the fraction of the conservation around the world that is being run by the 'white' Global North. My apologies if I have missed some key points here.

We do not intend to imply that most conservation work is being done by "white" countries, but rather that those are overwhelmingly the voices that are "heard" on a global stage. There is a lot of incredible work being done in both the conservation science and conservation practice sphere that is not run by the "white" Global North. However, we mainly refer to large "powerhouses" of conservation in this manuscript, which do tend to be situated in the Global North (presently and/or historically).

R2.8. Similar to the point directly above, I felt the paper at times discounted/ignored the great conservation work being done by the Asian Global North, as well as by local organizations in the Global South. E.g. lines 144-146; 229-235. As a personal anecdote, I recently spoke with the head of a Sri Lankan conservation NGO about the lack of racial diversity in 'western'

conservation/ecology/evolution. She is Sri Lankan, all of her staff are locals, and they are doing outstanding boots-on-the-ground conservation work on large mammals. According to her there are many people of colour working in conservation in South Asia. How would they fit into the narrative presented in this article? Similarly, one of my colleagues at a Canadian university is an economist of South-Asian descent, from India. He has done some really nice work on human-wildlife conflict in India with Indian academic institutions. Neither of these examples seem to fit into this paper's perspective on who is doing conservation work worldwide.

We thank the reviewers for this comment and wholeheartedly agree with their take on the great work being done by BIPOC people globally. Many authors on this manuscript would consider themselves a part of the communities of outstanding conservationists you refer to, and we do not mean to suggest that such people and their work doesn't exist. Instead, we are pointing to systems of power and influence that shape the entire landscape of global conservation. We believe that the vast in text edits we have made will help to make this distinction clearer.

R2.9. Personally, I would be more comfortable if the text could be altered slightly to emphasize the key problems associated with the type of conservation that is being done by academic institutions in the 'white' Global North. I don't think the paper is suggesting that all conservation done around the world is flawed, but readers might interpret it that way.

We agree and have made edits to address this as per our response to R2.8.

R2.10. Finally, I don't think that the authors have presented sufficient evidence to support the link that an increase in BIPOC representation in conservation science (in the 'white' Global North) will necessarily break that cycle between the twin spheres. I absolutely agree that diverse voices are needed, but if BIPOC entering this system are being judged and evaluated by the same 'old guard' of academia, how will BIPOC be able to break the cycle? I suppose we need to start somewhere don't we.

We are grateful for this comment and note that it echoes the well-known problem of recruitment overshadowing retention. We believe that rather than arguing explicitly that increasing representation in the 'white' Global North will break the cycle, we are stating that the suite of recommendations we present will do this. Improving recruitment of BIPOC individuals is just one part of the solution and done alone it will make little to no difference.

Line-by-line comments

R2.11. 39 – Black Lives Matter is not integrated at all in the text. We were not sure why it was included as a keyword/phrase. Also, did you mean to include 'equity' here and not 'equality'?

We have updated the keywords during this revision.

R2.12. 55- and throughout – both ‘equity’ and ‘equality’ are used throughout the paper. Please can the authors double check that they are using the right word in the right context?

Thank you for this comment, we have checked our use of both words throughout.

R2.13. 69-70- We assume this is written to speak to those who are unaware of these issues, so outlining some examples would be useful.

We have not included specific examples at this point in the text, but do provide examples later in the manuscript in the section “Conservation practice’s deep-seated racist history” (see response to R1.4 for further details).

R2.14. 81- Reference(s) required for the statement “In many places, mainstream conservation has replaced traditional resource management systems, often to the severe detriment of local people and biodiversity”

A reference has now been added (see line 109).

R2.15. 92- The paper uses words and phrases such as such as “leaches”, “vicious cycle”, “twin spheres”. These phrases are forceful and convey the importance and urgency of addressing racism and colonialism in conservation. We were a bit conflicted with this phrasing because on one hand the language is powerful and we cheered with fists raised, but on the other hand we worried that the very people who we need to read this paper would not be so enthusiastic. Tip-toeing around white fragility is infuriating but sadly sometimes a necessity.

We thank the reviewers for this comment. In several places we have reviewed the tone of language so as not to alienate potential allies. However, we have kept the use of “vicious cycle” and “twin spheres” as we believe these phrases most accurately describe the systems and processes we refer to.

R2.16. 97- and the surrounding paragraph - “we” is used both to mean the authors, and to mean the field at large. If possible please can you try to be more specific.

We thank R2 for drawing our attention to this and have amended appropriately (beginning at line 160).

R2.17. 99 – echoes Chaudhury and Colla 2020

We have added a citation for this paper (line 176).

R2.18. 113- Reference / example would be helpful to the reader for the claim: Contemporary conservation can perpetuate these values, often in spite of strenuous opposition from Indigenous and local people.

We have added a citation (line 212).

R2.19. 114-115 equates colonialism with racism – is this commonly accepted?

This has been edited so as not to directly equate the two.

R2.20. 116 - What definition of the Global North are you using?

We are grateful for this comment and have taken care to revise our use of terms throughout the manuscript as per our response to R2.6

R2.21. 116-117 It's not clear what this sentence means "Recognizing the extent of ecological degradation..." Who is doing the 'recognizing'? Also, what is 'true conservation'?

Thank you for highlighting the lack of clarity here- this sentence has now been revised.

R2.22. 120 – a definition of 'fortress conservation' would be helpful here

We have now added a definition to this part of the manuscript (line 220).

R2.23. 130 – E.g. Oxford, where many of the co-author currently work; perhaps this paper is a step towards reconciling past wrongs perpetuated by this institution? Many, many institutions still conduct neocolonial conservation work so this is not a dig at Oxford per se but we are mindful that some of the authors are from this institution.

We too recognise the atrocities, past and present, perpetuated by Oxford as an institution. However, we do not believe that this should prevent researchers from within this institution from speaking out about such things- quite the opposite. Throughout the manuscript we have referred to those needing to change in a way that is inclusive of the authors so as to be clear that we include ourselves in the broader community we are speaking to.

R2.24. 130- examples of the influential NGOs and research institutions would be helpful.

While we are grateful for this comment, we do not believe it is necessary for us to list the names of specific organisations/institutions. It would be difficult to include all of the names that likely should be included, and we believe that only referring to a subset would not be appropriate.

R2.25. 144-169 – these statements have been reviewed in several other comments/papers. It is unclear why they are listed here again; also, there are many outstanding BIPOC conservation practitioners and scholars (but maybe not necessarily working in western universities). I worry a bit that broad statements like 144 discount the contributions made by BIPOC conservation scholars.

We believe it is important to cover this content here (especially now with our expanded commentary post reviewers' comments) as we provide perspectives from backgrounds which have not yet been published in other papers. Please see our response to R2.8 for further comment on the statement in line 144, which has been reworded.

R2.26. 1-170 overall this section sometimes felt like a series of statements rather than a narrative that clearly explains how academia and institutional structures work together to produce graduates that have a partial / problematic conception of conservation, and how these structures continue to perpetuate inequalities in who is involved and at what level they are involved in conservation practice and decision making. To me the points made sense, but only because I had already read about these topics.

We are grateful for this comment and have made a series of structural changes and edits to the wording which we believe have addressed the problem.

R2.27. 172-184 – I agree that some degree of underrepresentation of BIPOC in conservation is due to racism. I would argue though, that the references you cite in this section do not necessarily show racism. In contrast, Milkman et al 2015 do show that faculty are more likely to respond to queries from white male students than from all other groups of students, although these findings are not specific to conservation (Journal of Applied Psychology 2015, 100:1678). There's a broader theme that permeates through the paper, and that is that underrepresentation is equated with racism. I agree that in many instances this is true, but some may argue that a lot of underrepresentation comes also from socioeconomic status (eg. aspects of Wanelik et al 2020).

We have added/changed many references in this section, as well as revised many of the statements made. We agree with the point about underrepresentation and socioeconomic status and have indicated this in the text when discussing the myriad of barriers to access that exist.

R2.28. 185 – do you know this to be a fact in institutions outside of the 'white' Global North?

We collectively agreed upon this statement across the authors, many of whom are not from the 'white' Global North.

R2.29. 196-209 – Pettorelli et al 2021 (How international journals can support ecology from the Global South), and Edwards et al 2018 (manuscripts from Asia were 5x more likely to be rejected) could be useful to bolster some of the statements presented here. The Pettorelli paper may be more relevant to conservation.

We thank the reviewers for drawing our attention to these important papers, but do not think they present facts that we are directly trying to convey at this point in the manuscript, so we have chosen not to include them.

R2.30. 229-231 – I mentioned this earlier but I'm wondering here about all of the conservation organizations in, for example, Pakistan, India, Philippines, Columbia, etc – are these organizations not doing meaningful work? Or maybe they are just really small compared to some of the large international organizations? This paragraph also seems to be a collection of important sentences that don't necessarily fit well together.

We have made several edits to this paragraph which we believe have improved the flow of the discussion. Please see earlier response to R2.7 and R2.8 for further details.

Figures –

R2.31. 1. Please define IPLC

We thank the reviewers for highlighting this- it has now been defined.

R2.32. 2 a,b These figures are beautifully illustrated. We would like a bit more explanation in each of the figure captions. All three of us struggled a bit with what exactly each part of each figure was depicting. For example, in the first diagram, what are the two people sitting at the table doing? And what is exactly happening on the roof of the 'grants' block?

We are grateful for this input and have taken care to elaborate further in the figure legend to fully explain the illustrations.

Appendix B

Rudd et al. Overcoming racism in the twin spheres of conservation Response to Referees

A version of the manuscript with revisions made since the previous submission can be found at the end of this document. Within this tracked changes version, comments referencing our responses can be found in the margin. All line numbers quoted in our responses below refer to the line numbers in this version of the manuscript.

Associate Editor

AE.1 I have now received two reviews from the original two referees. Reviewer one was very happy with the revisions, and only asks for some minor changes and edits. Reviewer 2 now sees less value in the paper, mainly from the perspective that the manuscript no longer reflects new thinking in the field. I did take this recommendation seriously. However, as associate editor, I believe that a scientifically sound paper still has substantial value even if other papers have come out on the topic in the meantime.

We gratefully received the second round of revisions from both reviewers and have responded to each below. We agree that other related papers have come out since ours was originally submitted, but we believe that we have approached the topic from a different angle, and as such have provided novel insights, perspectives and recommendations.

AE.2 This is why my recommendation is to focus on implementation of the changes asked for by reviewer 1. That said, please also address a bit more in detail how problems in conservation practice link to problems in conservation science, as asked for by reviewer 2.

We have implemented and addressed all the changes asked for by reviewer 1 as indicated below. We have also made significant additions to the manuscript to further tie the problems in conservation practice with those in conservation science. These edits can be found at lines 423 to 427. Additionally we have made significant improvements to the recommendations provided in boxes 1-3.

Referee 1

R1.1 Thank you for this revised manuscript, and the evident care that you took with the revisions. Figure 1 is much better integrated with the manuscript and I appreciate that statements are better supported with examples and references. I just have a few extra comments.

We are very grateful to reviewer 1 for their helpful comments and suggestions in round 1 of revisions. We are glad to hear that Figure 1 is now more clearly integrated within the manuscript, and that the addition of examples and references has resulted in our statements being better supported.

R1.2 Line 85 “in countries which are BIPOC majority and BIPOC minority.” -> regardless of whether the country is BIPOC majority and BIPOC minority.

This has been amended in text, line 383

R1.3 Line 101 extra space in “programmes , which”

This has been amended in text, line 310

R1.4 Line 158 “Influential researchers and advocacy groups based in the Global North stridently advocate for extending their conservation ideologies to vastly different socioecological and cultural contexts, with seemingly no regard for traditional practices or ethics in those locations [25]. These reckless prescriptions”...tone it down here, words like “stridently, seemingly, reckless” do not impart extra information to the sentence, are more rhetorical.

We thank reviewer 1 for highlighting this section, and have reworded it as appropriate, line 431

R1.5 Line 182 has an abrupt shift from representation exclusion of BIPOC people from conservation decision making to physical exclusion -and then financial exclusion- of BIPOC people from outdoor space. I think you need to make a smoother transition between these two types of exclusion (different, although stemming from same racist paradigms).

We have edited this section slightly to improve the transition between topics, line 463

R1.6 Line 256 extra period in “relate. . Racial”

This has been amended in text, line 548

R1.7 Line 270 I found “almost ‘Brahminical’ reverence” jarring, especially in the context of this essay. Do we really have to appropriate a Hindu term here? It’s as bad as people talking of scared cows. I’m not saying this because I support the hierarchy of the Hindu caste system but because I think we need to not cavalierly borrow references from BIPOC-dominated religions.

This term was introduced by a BIPOC Hindu (by birth) author and thus was not co-opted as such. It is meant to bring attention to an elite group within BIPOC, a point in fact that R2 suggests we have not addressed sufficiently. Moreover, it is also meant metaphorically (e.g. Boston Brahmins) and aims to criticise 'high caste' elitism in any culture. We have used single quotation marks around the term to signal that we are using the term deliberately, acknowledging its multiple meanings (line 564). We have also added more explicit information about the diverse backgrounds of authors at line 368, to further address this comment.

Referee: 2

R2.1 This is my second time reviewing this manuscript. The authors sort of addressed my concerns in the first round, mostly by way of addressing reviewer 1's concerns. My main concerns from round one included that I did not really see how this piece differed from similar articles that have been published in the last two years. Haelewaters et al 2021 (Ten simple rules for Global North researchers to stop perpetuating helicopter research in the Global South) has also recently been published in PLoS Computational Biology, and it offers concrete steps for north-south collaborations. Cronin et al 2021 (Anti-racist interventions to transform ecology, evolution and conservation biology departments) has also been published recently and the authors do cite this paper.

We thank the reviewer for highlighting further papers of relevance to ours and agree that several related papers have been published in recent years. We see this as a positive step towards addressing the problems we (and others) have outlined. As indicated, we do cite the Cronin et al paper, and have now added the Haelewaters et al paper as well. Given the vast diversity in experiences and perspectives afforded by the different authors on this manuscript, we believe our manuscript further adds to the conversation in these published papers.

R2.2 The main messages that I get from this manuscript are a) conservation has a deep-seated racist past; b) BIPOC people are excluded from conservation science and practice. In the response to reviewers, the authors write that "This paper adds further to the existing discourse by focusing specifically on both conservation science and practice. We identify problems that are unique to each of these spheres and the way in which they perpetuate one another, laid out against the context of conservations history." I've read the revised ms over a few times now and I am still having trouble linking problems in conservation practice to problems in conservation science. (Evidence for the conservation science perpetuating problems in conservation practice is nicely discussed in the paper.) Maybe this information is in lines 112-113?

We are grateful to reviewer 2 for highlighting their concern about a lack of information regarding the tie between problems in conservation practice and science. As they indicate, line 112-113 is an example of this, and we have also added explicit sentences at line 423 to address this further. In addition, we made a series of adjustments to the introduction and Figure 1 (both the figure itself and the figure legend) to address this concern in the last round of revisions. We are hopeful that with our additional edits in this round of revisions, we will have rectified the problem.

R2.3 Along these lines, the manuscript both talks about 'largely independent spheres' as well as 'vicious cycle'. Are the cycles strictly within the spheres or is there one cycle that goes from one sphere to the next... and the latter, how can the spheres be largely independent.

We suggest that while the spheres of conservation practice and research are seemingly independent of one another in terms of their processes and structures, there are cycles that exist between the two (e.g. conservation science research informing conservation practice

policy, and priorities established by conservation practice organisations fuelling conservation science research). Further and more detailed depictions of these cycles can be found in Figure 1.

R2.4 The majority of the paper reviews the different ways that BIPOC can be excluded from conservation-related courses or from accessing outdoor spaces and illustrates instances where BIPOC have been mistreated in academia or conservation practice. These statements have been published elsewhere and there is not much that is novel here.

While some of these sentiments have been published elsewhere (which we cite in this manuscript) there is novelty in presenting these statements together in one manuscript, with the context and recommendations we provide. We believe that collating this information into one manuscript, with the additional insights and perspectives offered by our collective authorship (which spans many fields of expertise, multiple nationalities and ethnicities) provides further novelty.

R2.5 I'm uncomfortable with the level of generalizations regarding BIPOC throughout the pages, and I'll push back a bit and say that there are many cultures where it's just not the cultural norm to go hiking – it's not that folks feel unwelcome, it's just not something that's done regularly.

We are grateful for this comment, which has led us to edit to the way we introduce this section (see line 474). We completely agree that the examples we present in this section (and throughout the manuscript) are not exhaustive, and they were never intended to be framed as such.

R2.6 As someone who is not a white Caucasian, personally I'm a bit tired of all of publications outlining mistreatment against BIPOC. I would have liked the paper more to focus on concrete solutions and less on repeating what's already been said.

We would like to respond by stating that we have provided 3 boxes with suggestions for solutions (24 in total), which we further expanded following the comments highlighting the need for this from the reviewer. While, as previously stated in response to R2.1 and R2.4, there have been other papers published on related topics in recent years, we do not see this as reason to stop writing about such topics. We do not believe that issues such as those being discussed in our manuscript will be resolved unless people continue to speak about them. In the wake of the events of summer 2020, many people and organisations expressed a commitment to doing better, but we believe this will only happen if we continue the momentum and push for change. We recognise that reading about such mistreatment can be triggering for BIPOC scholars, and as such we have taken care to signpost the content of the manuscript in the title and abstract.

R2.7 Maybe this paper will resonate more with non-BIPOC scholars.

We hope that the paper will resonate with both BIPOC and non-BIPOC scholars, as we expect that most people this manuscript will reach in journal format, will possess a degree of

privilege within the conservation sphere. Even as BIPOC scholars, we should be willing and able to recognise the privilege we possess and work towards solutions. This is particularly important in conservation, given we frequently interact and work with local communities.

R2.8 Solutions are presented in boxes 1-3 but they are fairly sweeping and high-level statements and not as helpful as some of the other solutions that have already been presented in the literature (e.g. Haelewaters et al 2021, or Cronin et al 2021, others). Also, decolonizing the curriculum is not the same as increasing BIPOC content into the curriculum.

We thank the reviewer for this comment and spent a significant portion of time reviewing and bolstering the recommendations we provide. For example, we have added suggested readings recommendations to broaden perspectives in the curriculum and removed the word “decolonizing” as we agree it was misplaced. We have tried to walk a line between providing suggestions that are not so broad as to be unhelpful, but not so narrow as to be ungeneralisable to each individual’s/organisation’s situation. We have amended the wording of the recommendation at line 645 to better represent the points we are conveying.

R2.9 I also have a really hard time with the overall sentiment of this paper that racism in conservation science and practice can be fixed by making the fields more inclusive to BIPOC scholars and practitioners. I know plenty of racist, culturally illiterate BIPOC folks.

We are grateful for this comment and agree wholeheartedly that the problems we outline will not be fixed simply by increasing the number of BIPOC scholars and practitioners within these fields, as being BIPOC does not guarantee someone is culturally literate or anti-racist. Racist hierarchies and processes operate within every society and at multiple levels, not simply at the global scale of colonial legacy. However, inclusivity is not necessarily the same as recruitment, and fostering greater inclusivity is something we believe it is important to strive for as it should go some way towards addressing many of the problems outlined. We have added a section of text expressing this sentiment at line 386. We also provide suggestions for this in box 3. Additionally, there are many more actions that need to be taken as we outline in the concluding sections of our manuscript, and specifically provide recommendations for in boxes 1-2.

R2.10 Given all of the papers published on similar topics in this area, I would think about what can you do with this paper that would affect the most change. I think publishing it as a regular paper in a journal will limit its utility. Who do you want to read this paper and why, and how do you get into in their hands.

We do believe that publishing this as a paper is one of the best ways to reach an academic audience. We will also communicate the content of the manuscript in other formats and across multiple platforms, so as to reach a wider audience.